# Myopathy associated LDB3 mutation causes Z-disc disassembly and protein aggregation through PKCα and TSC2-mTOR downregulation

Pankaj Pathak [1,3], Yotam Blech-Hermoni [1,3], Kalpana Subedi[1], Jessica Mpamugo[1], Charissa Obeng-Nyarko[1], Rachel Ohman[1], Ilda Molloy[1], Malcolm Kates [1], Jessica Hale[1], Stacey Stauffer[2], Shyam K. Sharan [2] & Ami Mankodi [1✉]

Mechanical stress induced by contractions constantly threatens the integrity of muscle Z-disc, a crucial force-bearing structure in striated muscle. The PDZ-LIM proteins have been proposed to function as adaptors in transducing mechanical signals to preserve the Z-disc structure, however the underlying mechanisms remain poorly understood. Here, we show that LDB3, a well-characterized striated muscle PDZ-LIM protein, modulates mechanical stress signaling through interactions with the mechanosensing domain in filamin C, its chaperone HSPA8, and PKCα in the Z-disc of skeletal muscle. Studies of $Ldb3^{Ala165Val/+}$ mice indicate that the myopathy-associated LDB3 p.Ala165Val mutation triggers early aggregation of filamin C and its chaperones at muscle Z-disc before aggregation of the mutant protein. The mutation causes protein aggregation and eventually Z-disc myofibrillar disruption by impairing PKCα and TSC2-mTOR, two important signaling pathways regulating protein stability and disposal of damaged cytoskeletal components at a major mechanosensor hub in the Z-disc of skeletal muscle.

[1] Neurogenetics Branch, National Institute of Neurological Disorders and Stroke, Bethesda, MD, USA. [2] Mouse Cancer Genetics Program, Center for Cancer Research, National Cancer Institute, Frederick, MD, USA. [3] These authors contributed equally: Pankaj Pathak, Yotam Blech-Hermoni. ✉email: Ami.Mankodi@nih.gov

Highly ordered arrangement of actin and myosin filaments in sarcomeres produces physical force in striated muscle. To this end, the Z-disc is of prime importance, anchoring actin filaments from adjoining sarcomeres. Striated muscle fibers are constantly exposed to strong mechanical stress. The Z-disc protein assemblies together with the dystrophin-associated glycoprotein complex and the integrin complex in the sarcolemma play essential roles in withstanding the extreme mechanical force generated during muscle contraction[1]. Genetic defects in the components of each of these large protein complexes lead to degenerative muscle diseases, indicating the importance of these interconnecting systems for the muscle integrity and function[2,3]. Mutations in genes encoding many of the Z-disc proteins have been found to cause myofibrillar myopathies (MFMs), which are characterized by primary dissolution of myofibrils near the Z-disc and accumulation of degraded proteins in the sarcoplasm of muscle fibers[4,5]. The dominant p.Ala165Val mutation in exon 6 of *LIM domain-binding 3* gene (*LDB3*; HGNC 15710; rs121908334; NM_001080114.2:c.494 C > T, NP_001073583.1:(p.Ala165Val)) has been reported in several unrelated families of European ancestry[6,7]. Whereas this mutation is interpreted as likely benign (https://varsome.com/variant/hg38/rs121908334), its penetrance in the long-studied Markesbery–Griggs pedigree is 100% by age 60 years and molecular studies of six unrelated families indicated a founder mutation with a common ancient ancestry[6]. Affected individuals present with adult-onset muscle weakness prominently affecting the calf muscles[6–8]. Cardiomyopathy is usually very late and only seen in a minority of patients, which may be due to extremely low expression of LDB3 isoforms containing the exon 6 in heart[9,10].

LDB3 (ZASP; Cypher) is a highly conserved PDZ-LIM protein that plays an essential but as yet undefined role in maintaining the Z-disc integrity in contracting muscle fibers in flies, zebrafish, and mice[11–13]. The PDZ domain interacts with structural Z-disc proteins and the LIM domains bind to and are phosphorylated by protein kinase C (PKC) isoforms[14,15]. Alternative splicing of *LDB3* is known to generate three major isoforms in skeletal muscle[9,10]. The shorter isoform lacks the C-terminal LIM domains. The longer isoforms with LIM domains either contain exon 10 or exclude it (LDB3-L and LDB3-LΔex10, respectively). LDB3-L is replaced by LDB3-LΔex10 in skeletal muscle during postnatal development[9]. Gene deletion studies in mice showed that the longer isoforms, but not LDB3-S, are important for Z-disc integrity in striated muscle[16]. Moreover, the p.Ala165Val mutation in LDB3-LΔex10, but not other isoforms, causes F-actin disruption in transfected muscle cells[17]. The actin-binding domain of LDB3 is mutated in MFM, but recombinant mutant proteins are correctly folded and show unaffected actin-binding affinity and kinetics[17,18]. Over-expression of the LDB3-LΔex10-p.Ala165Val via intramuscular injection leads to MFM-like pathology in mouse tibialis anterior muscle fibers[17]. However, short-term, variable, and heterogeneous expression of mutant protein in electroporated muscle fibers limited optimal investigation of disease mechanisms. Whereas knockout and cardiac-specific models have helped to characterize some LDB3 functions[11,16,19,20], the murine models of LDB3-MFM have not yet been reported and the LDB3 interactions relevant to the MFM phenotype remain unknown.

In this study, a heterozygous knock-in of the p.Ala165Val mutation in mouse *Ldb3* gene closely recapitulated the genetic mutation in patients and allowed physiological levels of LDB3 isoforms in mouse tissues. The *Ldb3^{Ala165Val/+}* mice developed muscle weakness and classic MFM pathology. Our results indicate that LDB3 acts as a signaling adapter in a major mechanosensor assembly through interactions with filamin C, its chaperone HSPA8, and PKCα at skeletal muscle Z-disc. The LDB3 p.Ala165Val mutation impairs PKCα and TSC2-mTOR mediated homeostasis in this large protein assembly leading to protein aggregation myopathy.

## Results

**Generation of *Ldb3^{Ala165Val/+}* knock-in mice.** We introduced the p.Ala165Val point mutation (chr14:34571772 C > T; GRCm38/mm10; C57BL/6 N), responsible for MFM[6,7], into exon 6 of the endogenous mouse *Ldb3* gene to generate *Ldb3^{Ala165Val/+}* mice by homologous recombination (Fig. 1a; Supplementary Fig. 1a). The mutated residue is conserved, and overall amino acid identity is >92% for LDB3 isoforms between human and mouse. Targeted gene sequencing and Southern blot analysis confirmed the accuracy of gene editing and the absence of other mutations in the recombined region (Supplementary Fig. 1b). The presence of the NP_001034164.1:(p.Ala165Val) mutation was further validated by Sanger sequencing (Fig. 1b). The levels of *Ldb3* mRNA and that of the major LDB3 protein isoforms in the vastus muscle of *Ldb3^{Ala165Val/+}* mice were similar to *Ldb3^{+/+}* littermates (Fig. 1c; Supplementary Fig. 1c; Supplementary Data 1), suggesting that the point mutation is not affecting transcript or protein stability in muscle tissue. Mating of *Ldb3^{Ala165Val/+}* mice resulted in 26% *Ldb3^{+/+}*, 54% *Ldb3^{Ala165Val/+}*, and 20% *Ldb3^{Ala165Val/Ala165Val}* mice ($n = 84$ mice; 10 litters), indicating that the mutation did not affect mouse viability. *Ldb3^{Ala165Val/+}* mice had a normal lifespan and weight. Their cage behavior, feeding, and grooming activities were comparable to *Ldb3^{+/+}* littermates.

***Ldb3^{Ala165Val/+}* mice develop progressive muscle weakness.** We screened *Ldb3^{Ala165Val/+}* mice and *Ldb3^{+/+}* littermates at 3, 6, and 9 months of age for signs of muscle weakness. The mice had normal locomotor coordination assessed by Rotarod. Grip strength tests showed a significant decline in the *Ldb3^{Ala165Val/+}* mice compared with *Ldb3^{+/+}* littermates at 3, 6, and 9 months of age (Fig. 1d; Supplementary Data 2). A two-way analysis of variance (ANOVA) with post hoc Bonferroni correction yielded significant effects of genotype, $F(1, 56) = 63.3$, $p < 0.0001$ and age, $F(2, 56) = 32.9$, $p < 0.0001$, as well as the age and genotype interaction, $F(2, 56) = 6.5$, $p = 0.003$. We found that the reduced grip strength was associated with decreased specific isometric force of the extensor digitorum longus muscle in 6-month-old *Ldb3^{Ala165Val/+}* mice compared with that in *Ldb3^{+/+}* littermates (Fig. 1e; Supplementary Data 3). A two-way ANOVA with post hoc Bonferroni correction yielded a significant effect of genotype $F(1, 5) = 17.93$, $p = 0.008$. Mean physical impulse to hold the wire grid reflecting the total sustained force exerted to oppose the gravitational force was decreased by about 60% in 9-month-old *Ldb3^{Ala165Val/+}* mice compared with that in *Ldb3^{+/+}* littermates (Fig. 1f; Supplementary Data 4).

**LDB3 p.Ala165Val mutation causes a classic pathological MFM phenotype in mouse skeletal muscle.** Skeletal muscle histology was assessed at 4, 6, and 8 months of age in the hindlimb muscles of *Ldb3^{Ala165Val/+}* mice and their *Ldb3^{+/+}* littermates. Muscle histology in mutant mice at 4 months of age was similar to that in wildtype mice (Fig. 2a). In contrast, Gomori trichrome staining of transverse sections of the soleus muscle of 8-month-old *Ldb3^{Ala165Val/+}* mice showed dark blue to blue-red hyaline granular deposits and rimmed vacuoles in the sarcoplasm of muscle fibers (Fig. 2a). These abnormal fibers showed decrease in oxidative enzyme activity in the region of protein deposits and increased enzyme activity mostly in the periphery, suggesting abnormal mitochondrial function. In addition, non-specific myopathic changes such as increases in internal nuclei, hypertrophic fibers, and muscle fibers with rounded contour were observed in the vastus and tibialis anterior muscles of mutant mice (Fig. 2b). To quantitate the changes in myonuclear location, we

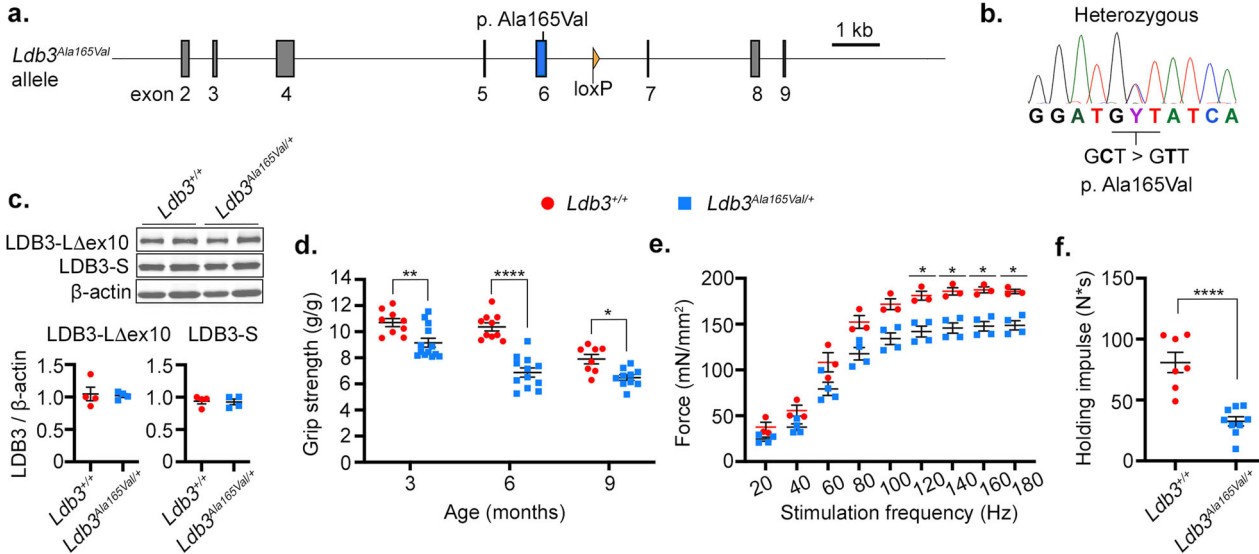

**Fig. 1 Generation and phenotyping of *Ldb3^Ala165Val/+* mice. a** Knock-in of the p.Ala165Val mutation in exon 6 (blue) of the mouse *Ldb3* gene. Residual loxP site post-*Cre* recombination is seen in intron 6 (orange triangle). **b** Sanger sequence shows the heterozygous C > T mutation changing the codon GCT (Ala) to GTT (Val). **c** Immunoblotting analysis (blot and dot plot) of the LDB3 isoforms expression relative to β−actin in the vastus muscle of 8-month-old *Ldb3^Ala165Val/+* mice (blue) and *Ldb3^+/+* littermates (red). n = 4 each, represents triplicate assay. **d** Dot plot comparing maximum all paws grip force of five pulls per mouse normalized to body weight (g/g) between *Ldb3^Ala165Val/+* mice (blue) and *Ldb3^+/+* littermates (red) at 3, 6, and 9 months of age. Mean ± SEM: 9.2 ± 0.3 g/g versus 10.7 ± 0.3 g/g at 3 months, 6.9 ± 0.3 g/g versus 10.4 ± 0.3 g/g at 6 months, and 6.5 ± 0.2 g/g versus 7.9 ± 0.3 g/g at 9 months; n = 10–13 *Ldb3^Ala165Val/+* mice and n = 8–10 *Ldb3^+/+* mice per age group. The two-way ANOVA Bonferroni's multiple comparisons test significant p values between groups are shown. **e** Dot plot shows maximal isometric force (mN/mm²) generated by the extensor digitorum longus muscle of 6-month-old male *Ldb3^Ala165Val/+* mice (blue) versus *Ldb3^+/+* littermates (red) against stimulation frequencies (Hz). Mean ± SEM: 148 ± 5 mN/mm² versus 187 ± 3 mN/mm²; 160 Hz; n = 4 *Ldb3^Ala165Val/+* mice and n = 3 *Ldb3^+/+* mice. The two-way ANOVA Bonferroni's multiple comparisons test significant p values between groups are shown. **f** Dot plot shows the maximum holding impulse (N*s) in the four-limb wire grid holding test in 9-month-old male *Ldb3^Ala165Val/+* mice and *Ldb3^+/+* littermates. Mean ± SEM = 32 ± 4 N*s versus 81 ± 8 N*s; n = 9 *Ldb3^Ala165Val/+* mice and n = 7 *Ldb3^+/+* mice. Male mice were used in these tests as female mice showed intra- and inter-group variability. The two-tailed unpaired t-test comparison significance between groups is shown. *p < 0.05, **p < 0.01, ****p < 0.0001. The error bars in dot plots represent Mean (SEM).

performed morphometry with wheat germ agglutinin (WGA) to outline muscle membrane and distinguish muscle nuclei. Relative to wildtype littermates, *Ldb3^Ala165Val/+* mice had a significantly higher percentage of fibers with internal nuclei in the tibialis anterior muscle (p < 0.01; Supplementary Fig. 2a). Muscle fiber type distribution was unchanged in the soleus and vastus muscles of *Ldb3^Ala165Val/+* mice (Supplementary Fig. 2b–d). Muscle fiber size distribution for types I and IIA in *Ldb3^Ala165Val/+* mice was shifted towards right compared to those from *Ldb3^+/+* mice, whereas distribution of type IIB fiber size was unchanged (Supplementary Fig. 2e). Mean minimal Feret's diameter for fiber types I and IIA was higher in *Ldb3^Ala165Val/+* mice relative to wildtype mice, but the change was not statistically significant in the Bonferroni's multiple comparison test (Supplementary Fig. 2f). Accordingly, variance coefficient of the minimal muscle fiber diameter was unchanged for all fiber types in *Ldb3^Ala165Val/+* mice (Supplementary Fig. 2g).

Immunostaining studies showed sarcoplasmic accumulations of the MFM-associated Z-disc proteins including LDB3, myotilin, desmin, filamin C, and αB-crystallin, as well as ubiquitin in same fibers of 8-month-old *Ldb3^Ala165Val/+* mice (Fig. 2c). Counts of a mean of 498 fibers in transverse soleus muscle section of five mutant mice each showed a mean of 64% and 44% muscle fibers contained filamin C and ubiquitin accumulations, respectively (Supplementary Fig. 3a). These protein accumulations occurred in multiple or diffuse form in muscle sarcoplasm. The abnormal fibers were distributed focally surrounded by muscle fibers without protein aggregation. Such protein aggregates were not observed in *Ldb3^+/+* mice. Electron microscopy studies showed the Z-disc myofibrillar disruption, dislocated enlarged mitochondria, and autophagic vacuoles in the soleus muscle fibers of 6-

month-old *Ldb3^Ala165Val/+* mice and normal Z-disc and sarcomere architecture in their *Ldb3^+/+* littermates (Fig. 2d–f). An in vivo "autophagy flux" assay[21] detected an increase in autophagy markers LC3-II and sequestosome 1 (p62/SQSTM1) following 3 days of treatment with colchicine in skeletal muscle of 4-month-old *Ldb3^Ala165Val/+* mice compared to *Ldb3^+/+* littermates (Fig. 2g–i; Supplementary Data 5). These findings demonstrate that *Ldb3^Ala165Val/+* mice develop typical MFM pathology characterized by the Z-disc streaming, sarcoplasmic protein aggregates, rimmed vacuoles, and increased lysosomal autophagy in skeletal muscle.

Cardiac ventricular muscle fibers of *Ldb3^Ala165Val/+* mice displayed normal histology at 9 months of age (Supplementary Fig. 4a, b), which correlates with clinical observations of cardiomyopathy not being a regular feature in patients known to have the same mutation[22]. A knockout-validated custom-made rabbit polyclonal LDB3ex6ab antibody detected LDB3 in the skeletal muscle of mice but not in cardiac muscle (Supplementary Fig. 5a, b), supporting previous observations that the LDB3 isoforms containing exon 6 are predominantly expressed in skeletal muscle in mice[9].

**LDB3^Ala165Val protein triggers early aggregation of filamin C and its chaperones, eventually leading to aggregation of the mutant protein and Z-disc myofibrillar disruption.** Patients usually present with advanced stages of myopathic disease at the time of biopsy, thereby limiting studies of early pathological features in the MFM. We examined the early cellular and molecular events that lead to the MFM pathology in *Ldb3^Ala165Val/+* mice. We found filamin C aggregates in muscle fibers that had normal

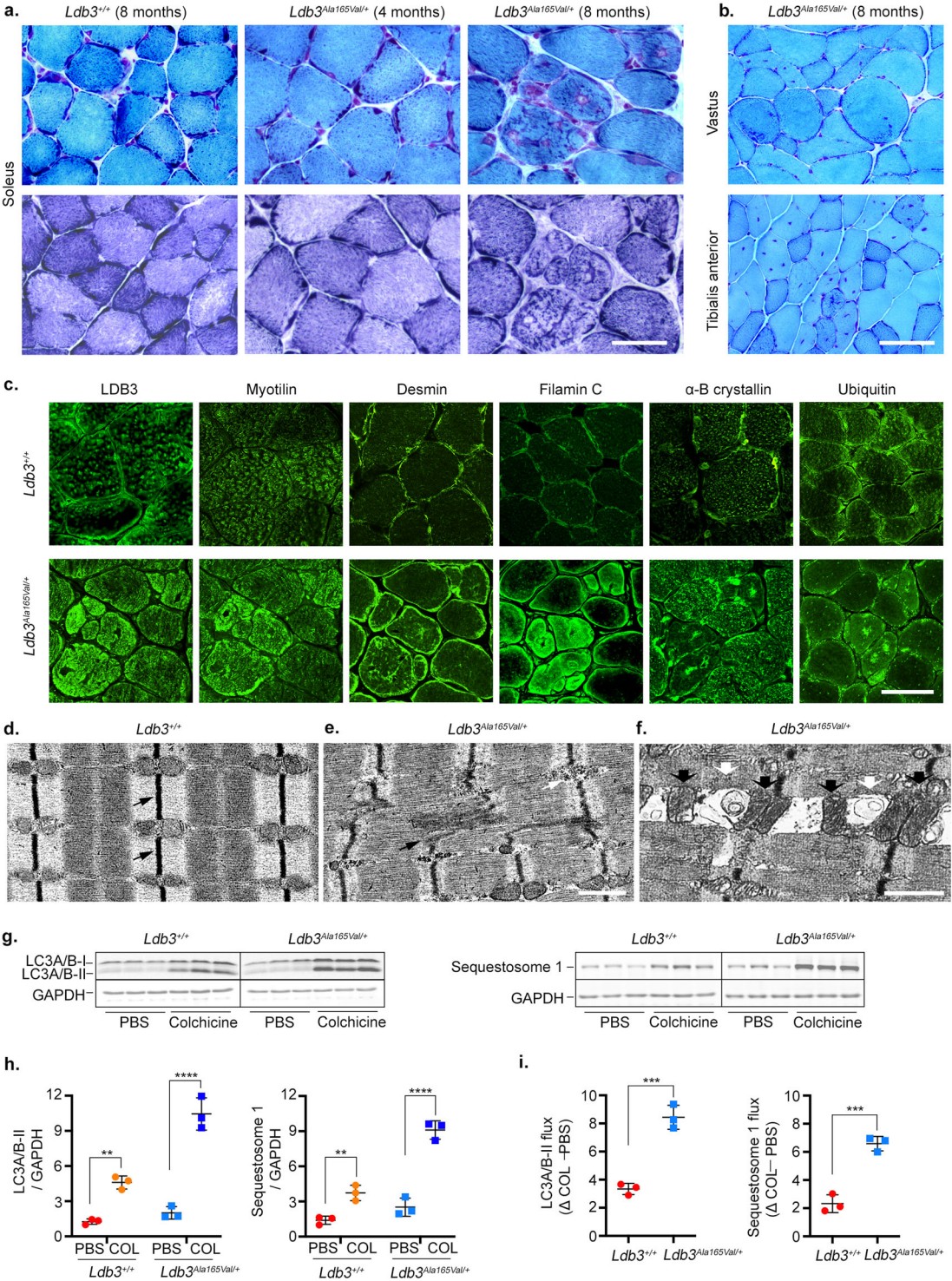

LDB3 immunostaining in transverse vastus and soleus muscle sections of 6-month-old $Ldb3^{Ala165Val/+}$ mice (Fig. 3a). Counts of a mean of 623 fibers from vastus muscle transverse sections of five mutant mice each showed a mean of 15% fibers contained filamin C accumulations (Supplementary Fig. 3b). Such protein aggregates were not seen in wildtype littermates. The filamin C accumulations were more prominent compared to LDB3 in the vastus lateralis muscle fibers of patients sharing the same mutation (Fig. 3a). Filamin C, an MFM gene product[23], acts as a primary mechanosensitive crosslink for actin filaments at the Z-discs[24]. Mechanical strain on the actin network results in unfolding of the

filamin C crosslinks, which are degraded through the conserved tension-induced chaperone-assisted selective autophagy (CASA) pathway[25]. Immunostaining of serial muscle transverse sections showed that filamin C aggregates were accompanied with the CASA chaperones BAG3, HSPA8, HSPB8, and ubiquitin in muscle fibers of $Ldb3^{Ala165Val/+}$ mice but not in their $Ldb3^{+/+}$ littermates (Fig. 3b, c). These sarcoplasmic protein aggregations were also found in the vastus muscle fibers of MFM patients with the LDB3 p.Ala165Val mutation (Fig. 3d).

Serial longitudinal tibialis anterior muscle sections of 4- and 8-month-old $Ldb3^{+/+}$ mice showed normal Z-disc distribution of

**Fig. 2 Muscle pathology of $Ldb3^{Ala165Val/+}$ mice. a** Representative Gomori trichrome (top; GT) and NADH-TR (bottom) stained adjacent frozen soleus muscle transverse sections of 4- and 8-month-old $Ldb3^{Ala165Val/+}$ mice ($n = 10$ and 16, respectively), and 8-month-old $Ldb3^{+/+}$ littermates ($n = 9$). **b** Representative images are GT – stained transverse frozen vastus and tibialis anterior muscle sections of 8-month-old $Ldb3^{Ala165Val/+}$ mice ($n = 16$). Muscle staining is normal in $Ldb3^{+/+}$ mice and 4-month-old $Ldb3^{Ala165Val/+}$ mice but shows sarcoplasmic aggregates and vacuoles (**a**), muscle fiber with rounded contour, hypertrophied fiber, and internal nuclei (**b**) in 8-month-old $Ldb3^{Ala165Val/+}$ mice. **c** Immunofluorescence staining of consecutive soleus muscle sections to those in (**a**) show sarcoplasmic protein accumulations in muscle fibers of 8-month-old $Ldb3^{Ala165Val/+}$ mice but not in $Ldb3^{+/+}$ mice. **d**–**f** Electron microscopy of the soleus muscle longitudinal section of 6-month-old $Ldb3^{+/+}$ mice ($n = 3$) shows normal Z-disc (black arrows) and the sarcomere architecture (**d**), whereas in the $Ldb3^{Ala165Val/+}$ littermates ($n = 3$) shows myofibrillar disruption at the Z-disc as streaming (black arrow; **e**) and accumulation of granular material (white arrow; **e**), as well as autophagic vacuoles (white arrow; **f**) and dislocated enlarged mitochondria (black arrow; **f**). **g, h** Immunoblotting analysis (blot and dot plot) of LC3A/B-II and sequestosome-1 (p62) protein levels, relative to GAPDH, in the vastus muscle of 4-month-old $Ldb3^{Ala165Val/+}$ and $Ldb3^{+/+}$ littermates after three days of colchicine (COL) or PBS treatment. $n = 3$ mice per treatment group and triplicate assays. **i** Dot plot of LC3A/B-II and sequestosome-1 flux [$\Delta$COL − PBS]. Mean ± SD flux: LC3A/B-II: 8.4 ± 0.8 versus 3.3 ± 0.4 and sequestosome-1: 6.6 ± 0.5 versus 2.3 ± 0.6 in $Ldb3^{Ala165Val/+}$ mice and $Ldb3^{+/+}$ littermates, respectively. The two-way ANOVA Bonferroni's multiple comparisons test (**h**) and the two-tailed unpaired $t$-test comparison (**i**) significant $p$ values are shown. $**p < 0.01$; $***p < 0.001$; $****p < 0.0001$. The error bars in dot plots represent Mean (SD). See Supplementary Fig. 6b for full-length blots. Scale bars = 50 µm (**a**–**c**), 1 µm (**d**–**e**), and 2 µm (**f**).

LDB3, filamin C, myotilin, BAG3, and HSPA8 proteins (Fig. 4a and f). Immunostaining of the tibialis anterior muscle sections in 4-month-old $Ldb3^{Ala165Val/+}$ mice showed aggregates of filamin C, BAG3, and HSPA8 co-localizing at the Z-discs of same muscle fibers, whereas staining for LDB3 and myotilin was normal (Fig. 4b–e). Co-aggregation of LDB3 with filamin C, myotilin, BAG3, and HSPA8 with or without Z-disc myofibrillar disruption occurred at a later disease stage in 8-month-old $Ldb3^{Ala165Val/+}$ mice (Fig. 4g–j). Prominent accumulations of the CASA chaperones were found in muscle fibers harboring filamin C and other myofibrillar protein aggregates but not in adjoining fibers without protein aggregation (Figs. 3c, d and 4d, e, i, j). These results indicate that the LDB3 p.Ala165Val mutation causes aggregation of damaged filamin C before LDB3 and other MFM proteins, eventually leading to Z-disc myofibrillar disruption in skeletal muscle. Muscle fibers react to ubiquitinated protein aggregation with accumulations of the CASA chaperones, but the protein degradation systems are not efficient in clearing the damaged protein aggregates.

**LDB3$^{WT}$ and LDB3$^{Ala165Val}$ proteins interact with mechanosensing domain in filamin C and HSPA8, a chaperone in tension-induced autophagy pathway.** We found early and prominent filamin C aggregates in skeletal muscle fibers of $Ldb3^{Ala165Val/+}$ mice, indicating that the LDB3 p.Ala165Val mutation likely affects the stability or degradation of filamin C. Immunoblotting studies showed that whole muscle levels of filamin C and the CASA chaperones were not altered in skeletal muscle of $Ldb3^{Ala165Val/+}$ mice (Fig. 5a). The association of modular protein-interacting domains enables LDB3 to serve as an adapter protein that recruits multiple proteins to a localized site of action in a subcellular domain. To this end, we conducted a yeast two-hybrid (Y2H) screen of human skeletal muscle cDNA library using a human LDB3 peptide encoded by exons 8 and 11 as bait to identify novel interacting proteins. This bait region is exclusively present in the postnatal LDB3-LΔex10 isoform. The region is known to interact with skeletal actin, and its presence is crucial for the deleterious effects of the p. Aal165Val mutation on actin cytoskeleton in transfected muscle cells[17]. The Y2H screen identified 46 prey clones encoding six potential LDB3 interactors including two Z-disc associated proteins filamin C and HSPA8, with strong confidence in the interaction (Fig. 5b; Supplementary Table 1).

Interestingly, the five filamin C clones encoded the rod domain 2 immunoglobulin-like (Ig) 17–21 repeats, a precisely tuned mechanosensor that detects and responds to the small cytoskeletal forces during muscle contraction[26,27]. In addition, three prey clones encoded an internal region of HSPA8 (aa 237–468;

NM_006597.5) that is recruited by BAG3 to the Ig19–21 repeats of unfolded filamin C as a part of the CASA complex forming the molecular basis for sensing force-induced unfolding[25]. Pairwise Y2H assays showed that inclusion of the exon 10-encoded region in the bait corresponding to LDB3-L isoform abolished LDB3 binding to HSPA8 (Fig. 5c). These findings support our previous observations of isoform-specific LDB3 interactions with skeletal actin[17,18]. GST pulldown assays using tagged wildtype (WT) and mutant LDB3-LΔex10 proteins as bait showed that the LDB3 p. Ala165Val mutation did not impair filamin C and BAG3 interactions (Fig. 5d). Binding of WT and mutant LDB3-LΔex10 with filamin C Ig 17–21 domain and HSPA8 was validated by independent co-immunoprecipitation (IP) of tagged proteins from transfected COS-7 cell lysates (Fig. 5e). Endogenous interactions of LDB3 with filamin C and HSPA8 in skeletal muscle were confirmed by co-IP of these proteins from the tibialis anterior muscle myofibrillar fraction of 4-month-old $Ldb3^{Ala165Val/+}$ mice and $Ldb3^{+/+}$ littermates (Fig. 5f). These findings point to mechanosensing roles of LDB3 through interactions with filamin C and HSPA8 in the Z-disc. The p.Ala165Val mutation does not appear to affect these LDB3 interactions in skeletal muscle.

**LDB3$^{Ala165Val}$ protein impairs PKCα and TSC2-mTOR signaling in skeletal muscle.** Previous studies suggested that LDB3 can act as a signaling scaffold to regulate protein function[14,16,28–31]. To identify altered signaling pathways leading to MFM pathology in $Ldb3^{Ala165Val/+}$ mice, a reverse-phase protein array (RPPA)[32] was done on the vastus muscle lysates of 4- and 8-month-old $Ldb3^{Ala165Val/+}$ mice and their gender-matched $Ldb3^{+/+}$ littermates. It should be noted that filamin C aggregates were observed in skeletal muscle fibers of 4-month-old mutant mice (Fig. 4b, d), whereas Z-disc myofibrillar disruption is evident after 6 months of age (Figs. 2e and 4g–i). RPPA measured changes in the abundance and post-translational modifications of proteins using 248 antibodies. The analysis showed a significant downregulation of the LDB3 interactor PKCα[14], as well as the most important negative mTOR regulator TSC2[33] in the vastus muscle of 4-month-old $Ldb3^{Ala165Val/+}$ mice ($\geq$1.5-fold, corrected $p \leq$ 0.05; Fig. 6a; Supplementary Table 2a; Supplementary Data 6). Levels of PKCα and TSC2 were downregulated by 1.7 and 1.5-fold, respectively in the vastus muscle of 4-month-old $Ldb3^{Ala165Val/+}$ mice compared to $Ldb3^{+/+}$ littermates. PKCα levels were also downregulated by 1.8 fold in the vastus muscle of 8-month-old mutant mice relative to wildtype littermates (Fig. 6b; Supplementary Table 2b; Supplementary Data 7). TSC2 showed a downward trend in the older mutant mice (1.4 fold, corrected $p =$ 0.07). Validation of the RPPA results was done by immunoblotting of the vastus muscle lysates of $Ldb3^{Ala165Val/+}$ mice and

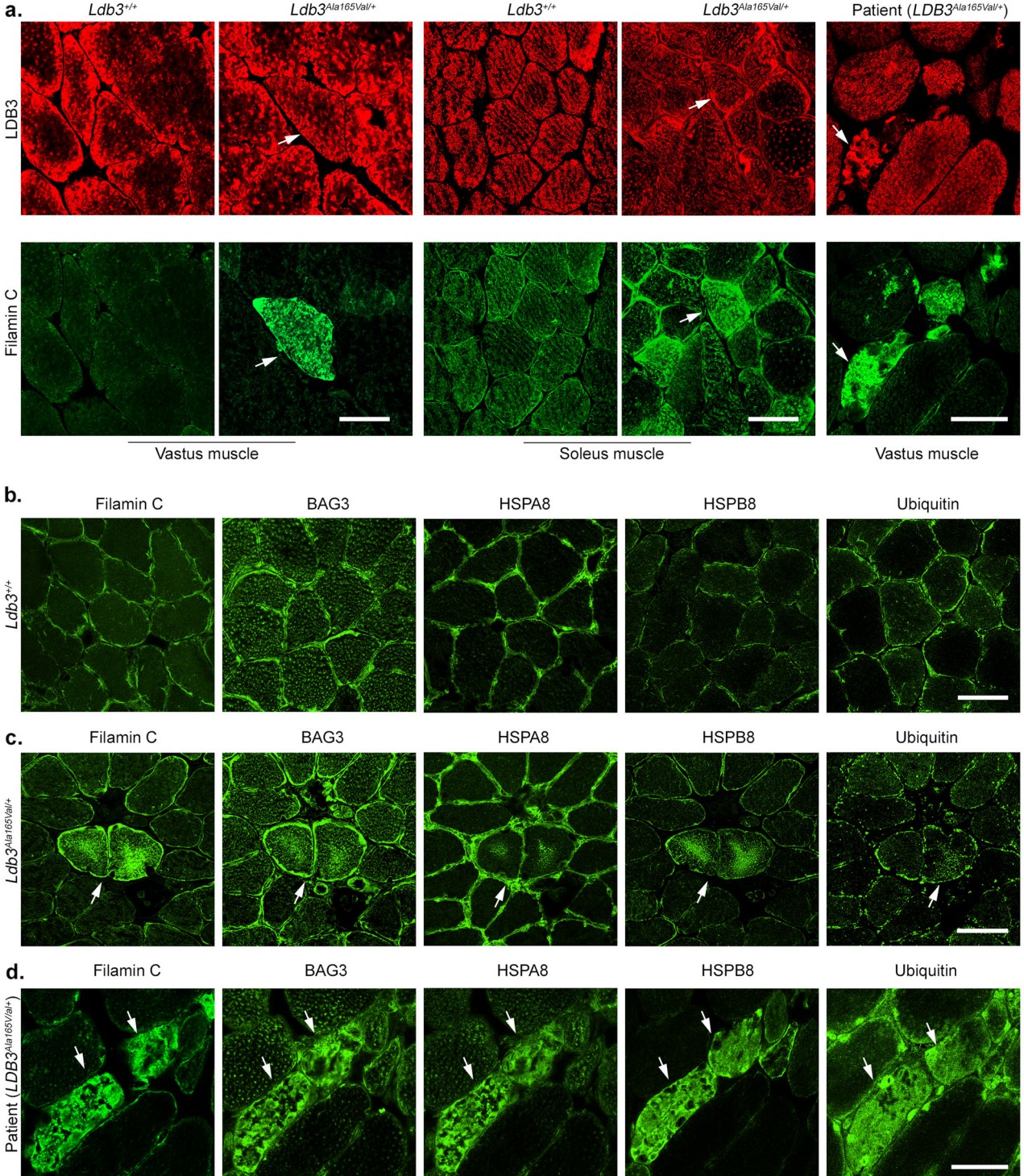

**Fig. 3 Immunolocalization of filamin C and CASA chaperone complex in skeletal muscle of *Ldb3^Ala165Val/+* mice and patients sharing the same mutation. a** Representative immunofluorescence on frozen vastus lateralis and soleus muscle transverse section of 6-month-old *Ldb3^Ala165Val/+* mice (*n* = 9 and 5, respectively) and *Ldb3^+/+* littermates (*n* = 5) and an MFM patient (*n* = 3) with the LDB3 p.Ala165Val mutation stained with filamin C and LDB3 antibodies. White arrows indicate muscle fibers with sarcoplasmic filamin C aggregates that have normal sarcoplasmic LDB3 distribution in *Ldb3^Ala165Val/+* mice and relatively less extensive LDB3 accumulations in the patient. **b–d** Representative immunofluorescence on frozen soleus muscle serial transverse sections of 8-month-old *Ldb3^+/+* mice (**b**; *n* = 6) and *Ldb3^Ala165Val/+* littermates (**c**; *n* = 8), and the vastus lateralis muscle of patient (**d**; *n* = 3) stained with filamin C, BAG3, HSPA8, HSPB8, and ubiquitin antibodies. White arrows indicate same muscle fibers with sarcoplasmic accumulations of filamin C and the CASA proteins in *Ldb3^Ala165Val/+* mice and patient. Such protein aggregates are not seen in the sections obtained from *Ldb3^+/+* mice. Scale bars = 50 μm.

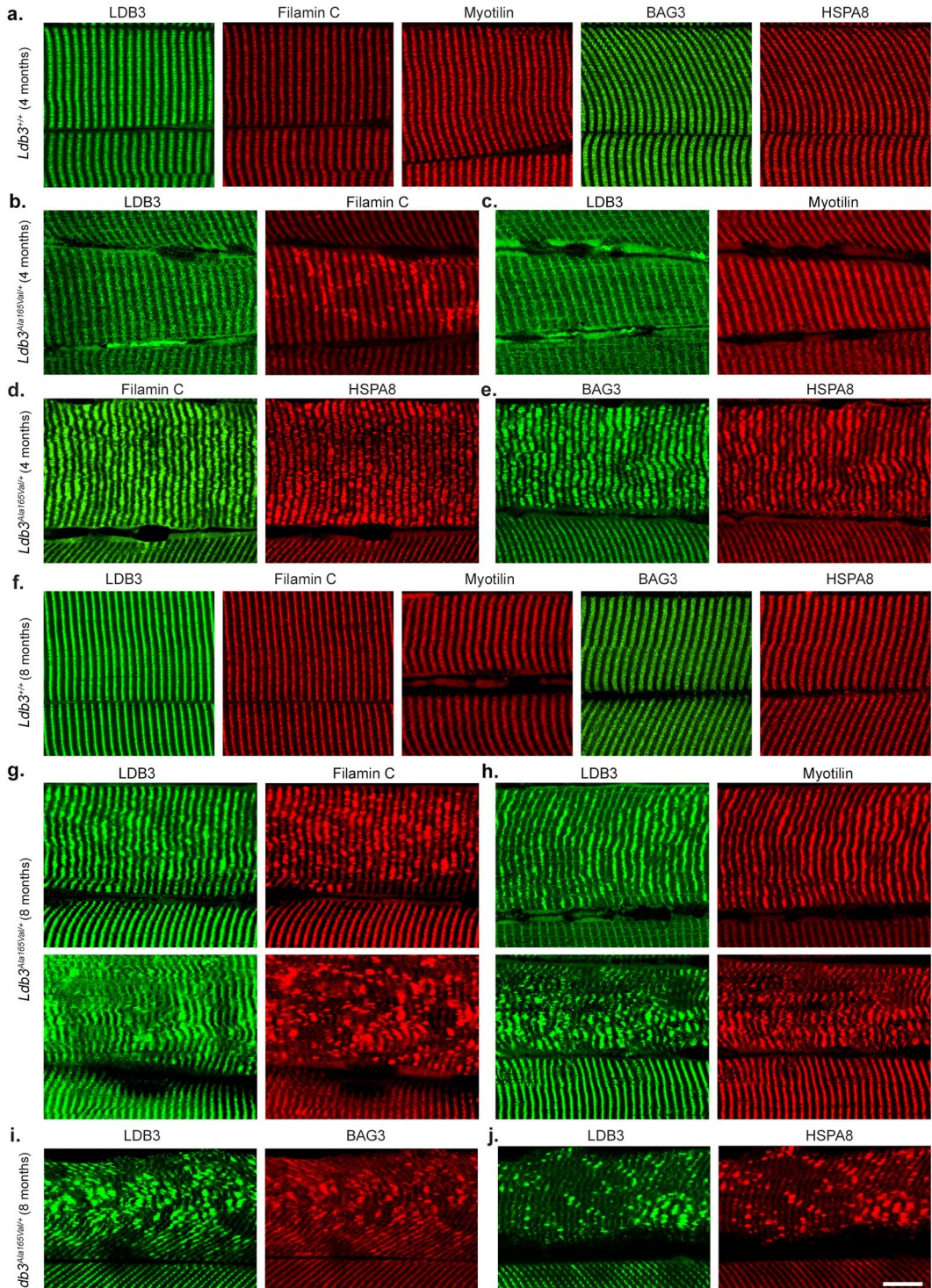

*Ldb3*<sup>+/+</sup> littermates. Whole muscle levels of PKCα in 4- and 8-month-old *Ldb3*<sup>Ala165Val/+</sup> mice were decreased to about 50% of those in *Ldb3*<sup>+/+</sup> littermates, and levels of TSC2 in the mutant mice were reduced to almost two-thirds of those in *Ldb3*<sup>+/+</sup> mice (Fig. 6c–f; Supplementary Data 8, 9). Downregulation of PKCα and TSC2 indicates a likely mechanism for the observed aggregation of damaged filamin C in muscle fibers of *Ldb3*<sup>Ala165Val/+</sup> mice as PKCα stabilizes filamin C in muscle Z-disc through phosphorylation[34], and TSC2 initiates CASA-mediated degradation of damaged filamin through local mTORC1 inhibition[35].

## Discussion

Here, we unveil a previously undescribed mechanism by which a disease-associated mutation in LDB3, a PDZ-LIM protein, causes protein aggregation and myofibrillar Z-disc disruption in skeletal muscle. Our data demonstrate that LDB3 directly binds to the mechanosensing domain of filamin C, and its chaperone HSPA8, a central player in protein folding and proteastasis control. Studies in knock-in mice show that the MFM-associated LDB3 p.Ala165Val mutation causes early aggregation of filamin C and its chaperones in the tension-induced CASA pathway before

**Fig. 4 Progressive protein aggregation and Z-disc myofibrillar disruption in skeletal muscle of *Ldb3^Ala165Val/+* mice.** Representative immunofluorescence staining on perfused tibialis anterior muscle consecutive longitudinal sections of 4-month-old (**a–e**) and 8-month-old (**f–j**) *Ldb3^Ala165Val/+* mice and their *Ldb3^+/+* littermates. **a** Muscle sections of 4-month-old *Ldb3^+/+* mice (*n* = 4) show normal Z-disc staining for LDB3, filamin C, myotilin, BAG3, and HSPA8 proteins. **b, c** Muscle sections of 4-month-old *Ldb3^Ala165Val/+* mice (*n* = 6) show same muscle fiber co-stained with LDB3 and filamin C antibodies (**b**), and LDB3 and myotilin antibodies (**c**). **d, e** Muscle sections of 4-month-old *Ldb3^Ala165Val/+* mice (*n* = 6) show same muscle fiber co-stained with filamin C and HSPA8 antibodies (**d**), and BAG3 and HSPA8 antibodies (**e**). Filamin C, HSPA8, and BAG3 aggregates are seen at the Z-disc spanning multiple sarcomeres in same fiber. In contrast, the LDB3 and myotilin antibodies show normal Z-disc staining in the fiber with filamin C aggregates. **f** Muscle sections of 8-month-old *Ldb3^+/+* mice (*n* = 5) show normal Z-disc staining for LDB3, filamin C, myotilin, BAG3, and HSPA8 proteins. Top and bottom panel each shows same muscle fiber co-stained with LDB3 and filamin C antibodies (**g**), and LDB3 and myotilin antibodies (**h**) in muscle sections of 8-month-old *Ldb3^Ala165Val/+* mice (*n* = 7). Muscle sections of 8-month-old *Ldb3^Ala165Val/+* mice (*n* = 7) show same muscle fiber co-stained with LDB3 and BAG3 antibodies (**i**), and LDB3 and HSPA8 antibodies (**j**). LDB3 aggregates colocalize with filamin C, myotilin, BAG3, and HSPA8 at the Z-discs in muscle fibers. The protein aggregates are seen in muscle fibers with normal periodicity of the Z-discs and with disorganized Z-disc periodicity indicating Z-disc myofibrillar disruption. A part of adjacent muscle fiber with normal staining pattern is shown at bottom in each image for comparison. Scale bar = 10 μm and applies to all images.

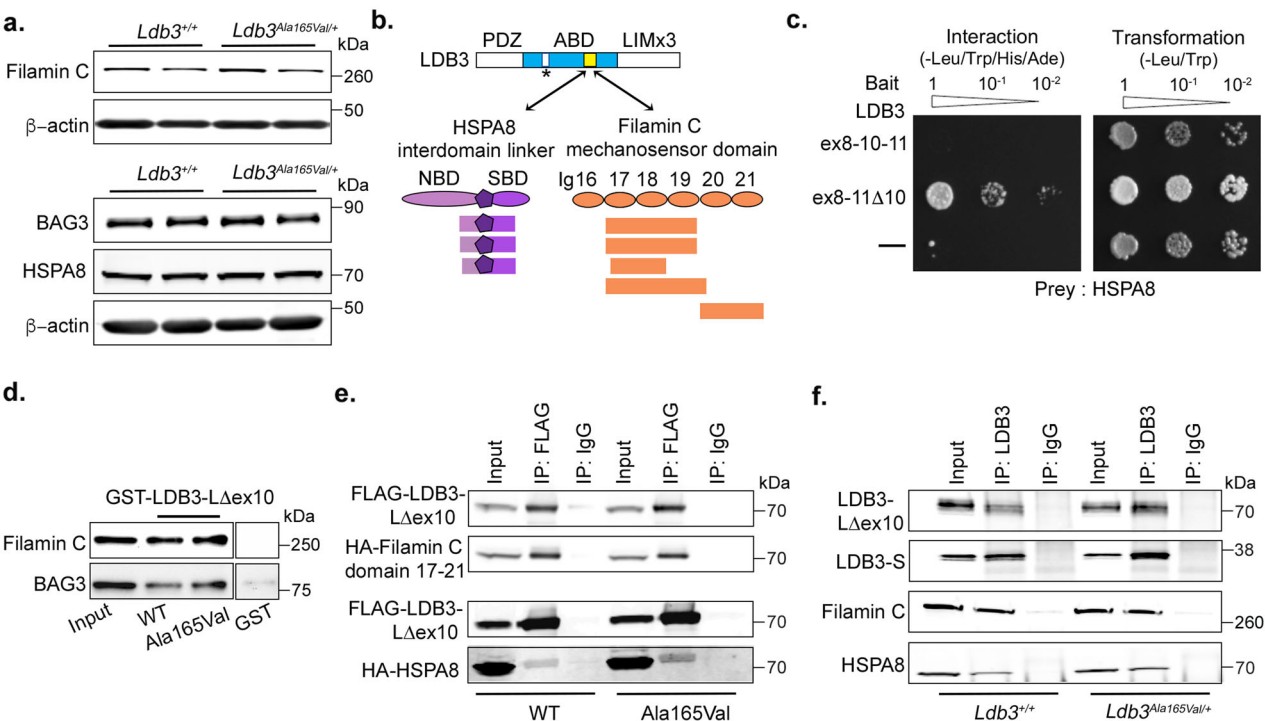

**Fig. 5 Characterization of LDB3 interactions with filamin C and HSPA8. a** Representative immunoblots showing protein levels of filamin C, BAG3, and HSPA8 relative to *β*-actin in the vastus muscle of 8-month-old *Ldb3^Ala165Val/+* mice and *Ldb3^+/+* littermates. Data represent *n* = 5 mice per group and triplicate assays. **b** Schematics of LDB3 interaction with the mechanosensing domains Ig17–21 of filamin C and its chaperone HSPA8 as identified by yeast two-hybrid (Y2H) screen of a human skeletal muscle cDNA library. See Supplementary Table 1. Locations of the LDB3 bait encoded by exons 8-11Δ10 (yellow) and the p.Ala165Val mutation (white; asterisk) within actin-binding domain (ABD; blue) are shown in LDB3-LΔex10 isoform[7,17]. Domain composition of the prey clones are shown. **c** Pairwise Y2H assays demonstrating interaction between LDB3 peptides and HSPA8. Positive interactions show yeast growth on the media deficient in *HIS3* and *ADE2*. Yeast cells co-transformed with empty bait vector and the HSPA8 prey show no growth (labeled –). Transformation efficiency was uniform for all constructs. Sequential tenfold yeast dilutions are shown. **d** GST pulldown assay shows that GST-tagged wildtype (WT) and mutant LDB3-LΔEX10 (Ala165Val) but not GST alone pulled down filamin C and its interactor the CASA cochaperone BAG3 from the vastus muscle lysates of wildtype mice. Data represent *n* = 3 mice and triplicate assays. **e** Co-immunoprecipitation (co-IP) assays show that a FLAG antibody pulled down the FLAG-tagged WT and mutant (Ala165Val) LDB3-LΔex10 together with HA-tagged filamin C rod domain Ig17–21 and HSPA8 in Cos7 cells. Data represent triplicate assays. The proteins were detected with anti-FLAG and anti-HA antibodies. **f** Co-IP assays show that an LDB3 antibody pulled down LDB3 isoforms together with filamin C and HSPA8 from the tibialis anterior muscle lysates of *Ldb3^Ala165Val/+* and *Ldb3^+/+* mice. Data represent *n* = 3 mice per group and triplicate assays.

aggregation of the mutant LDB3 protein in muscle fibers. The LDB3 p.Ala165Val mutation destabilizes filamin C and stalls the removal of damaged filamin C through impaired PKCα and TSC2-mTOR signaling at arguably the most important mechanosensor hub in the Z-disc of skeletal muscle (summarized in Fig. 7). The presence of chaperones and ubiquitin in these protein aggregates likely represents impairment of protein degradation

pathways. Further, pathology studies show that muscle fibers with protein aggregates develop the Z-disc myofibrillar disruption in skeletal muscle of *Ldb3^Ala165Val/+* mice.

LDB3 is well-poised to play a role in mechanosensor function in striated muscle. Homozygous LDB3-null mice display fragmented Z-discs in the diaphragm muscle after birth, but not during embryonic development[11]. Since the diaphragm does not

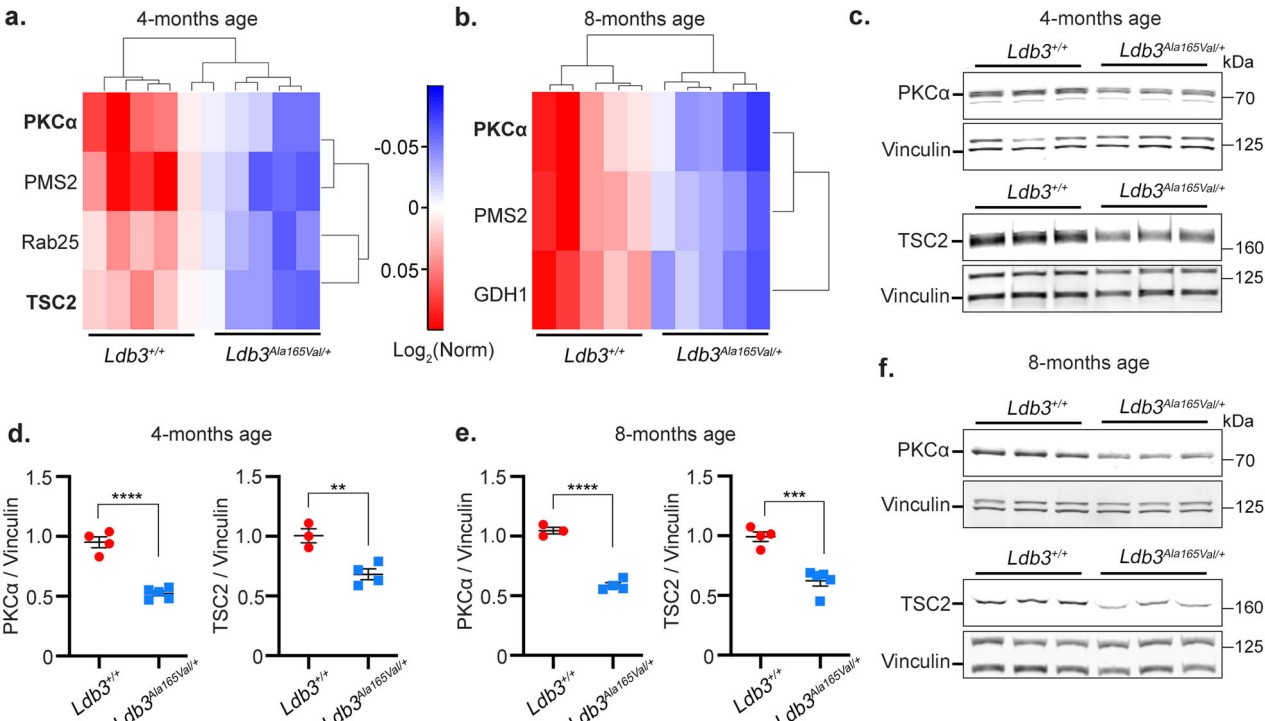

**Fig. 6 Effects of LDB3 p.Ala165Val mutation on PKCα and TSC2-mTOR expression in skeletal muscle of *Ldb3^Ala165Val/+* mice.** Heatmap of protein levels in the vastus muscle lysates of 4 months (**a**) and 8-month-old (**b**) *Ldb3^Ala165Val/+* mice and their gender-matched *Ldb3^+/+* littermates (n = 5 per group; total 20 mice) detected by RPPA. Differentially expressed proteins with statistically significant fold changes are shown on the vertical axis (≥1.5-fold, corrected $p \leq 0.05$). The levels of each protein are presented with colors, with blue for lowest and red for highest. See Supplementary Table 2. Only PKCα and TSC2 are known to localize at skeletal muscle Z-disc. **c, d** Immunoblotting analysis (blot and dot plot) of PKCα and TSC2 protein levels relative to vinculin in the vastus muscle of 4-month-old *Ldb3^Ala165Val/+* mice (n = 5 and 4, respectively) and *Ldb3^+/+* littermates (n = 4 and 3, respectively). **e, f** Immunoblotting analysis (blot and dot plot) for protein levels of PKCα and TSC2 relative to vinculin in the vastus muscle of 8-month-old *Ldb3^Ala165Val/+* mice (n = 4 and 5, respectively) and *Ldb3^+/+* littermates (n = 3 and 4, respectively). The error bars in dot plots are Mean (SEM) and represent triplicate assays. The two-tailed unpaired t-test comparison significant p values are shown. **$p < 0.01$; ***$p < 0.001$; ****$p < 0.0001$.

contract until after birth, these observations suggest that LDB3 is required for maintaining the sarcomere structure during muscle contraction. LDB3 interacts with skeletal actin filaments, the actin crosslinkers α-actinin and myotilin, as well as integrins that play a role in sensing external tension in striated muscle[13–15,17,18]. In this study, we identified an LDB3-binding site in filamin C rod domain 2 that is involved in multiple protein interactions including BAG3, myotilin, dystroglycans, and integrins (Fig. 7a)[25,36,37]. Whereas the mechanical properties of filamin C remain unknown, previous studies of filamin A showed that the rod 2 domain undergoes unfolding upon mechanical strain that changes affinity of the binding proteins, thus providing a molecular basis for mechanosensing[24,26,27]. LDB3 also binds to HSPA8, which together with BAG3 degrades unfolded filamin C through the CASA pathway upon mechanical strain (Fig. 7b)[25]. These interactions were unaffected by the MFM-associated LDB3 p.Ala165Val mutation, supporting that toxicity is not through altered ligand binding properties of mutant protein. The filamin C- and HSPA8-binding domain in LDB3 also interacts with skeletal actin and is uniquely present in the postnatal longer LDB3 isoform[17]. Previously we have shown that the LDB3 p. Ala165Val mutation in the postnatal longer LDB3 isoform, but not other isoforms, leads to myofibrillar disruption[17], suggesting that the LDB3 interactions through this domain likely play essential roles in the pathogenesis of MFM. This is supported by early and prominent aggregation of filamin C that seeds aggregation of other proteins in muscle fibers of *Ldb3^Ala165Val/+* mice. Mutations in many proteins in this interactome, such as LDB3, filamin C, myotilin, BAG3, DNAJB6, and HSPB8 lead to MFM

phenotype[5], indicating that most of the disease pathways converge onto a final pathway centered on the large LDB3-filamin C-CASA chaperone scaffold for this diverse group of myopathies. These findings underscore critical roles for LDB3 in recruiting key proteins to strategic sites and facilitating mechanosensing in striated muscle.

LDB3 is known to anchor PKC isozymes at the Z-disc of striated muscle[14]. PKCα accounts for 97% of the conventional PKC function in skeletal muscle[38]. We found a significant decrease (~50%) in PKCα expression in skeletal muscle of *Ldb3^Ala165Val/+* mice, before the MFM phenotype. Mechanisms by which the MFM-associated LDB3 p.Ala165Val mutation downregulates PKCα in skeletal muscle are yet unknown, but presumably the mutation affects ligand binding, phosphorylation, or autoinhibition of the kinase that effectively leads to its degradation[39,40]. The longer LDB3 isoforms have been shown to negatively regulate the PKCα expression in heart[16]. To this end, the p.Ala165Val mutation likely acts as a gain-of-function mutation leading to the MFM phenotype. Interestingly, PKCα has been viewed as an attractive therapeutic target in many diverse cancers and degenerative diseases of heart and brain[39–41]. Loss-of-function mutations in PKCα are prevalent in cancers, and studies in cancer cell lines showed that small changes in the PKC expression have large effects on cellular functions[40]. It is known that PKCα phosphorylates filamin C and prevents its calpain-mediated proteolysis, thus preserving its dimerization function[34]. An MFM-associated nonsense mutation in filamin C, which disrupts its dimerization domain, causes self-aggregation of mutant protein and leads to aggregation of interacting

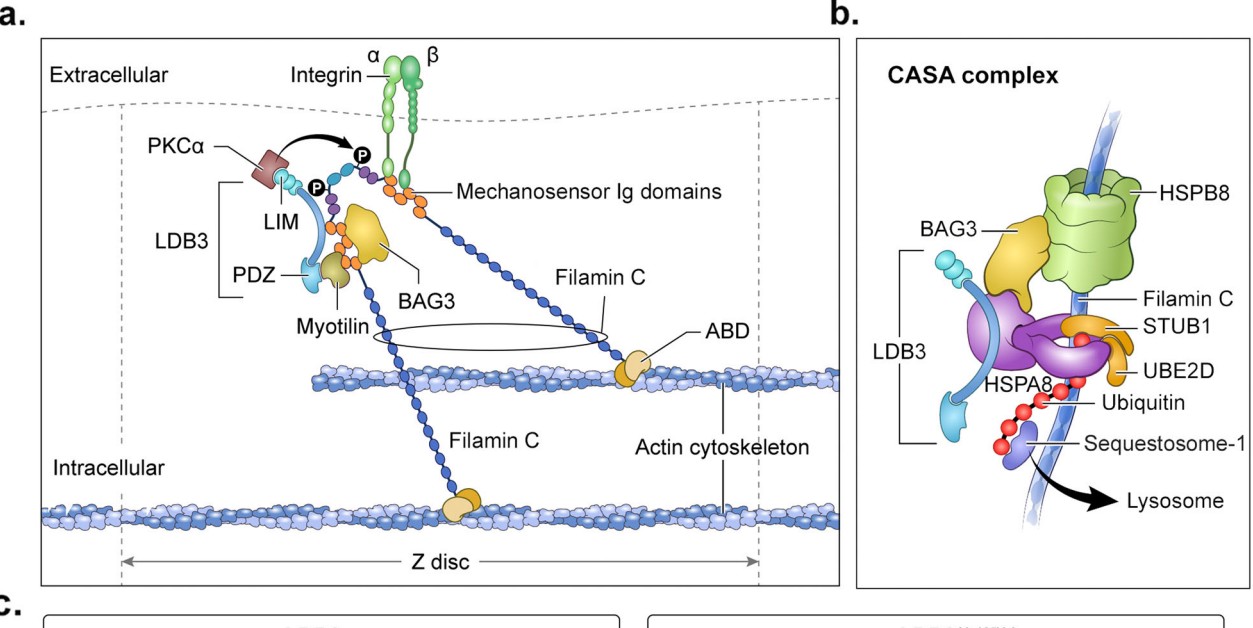

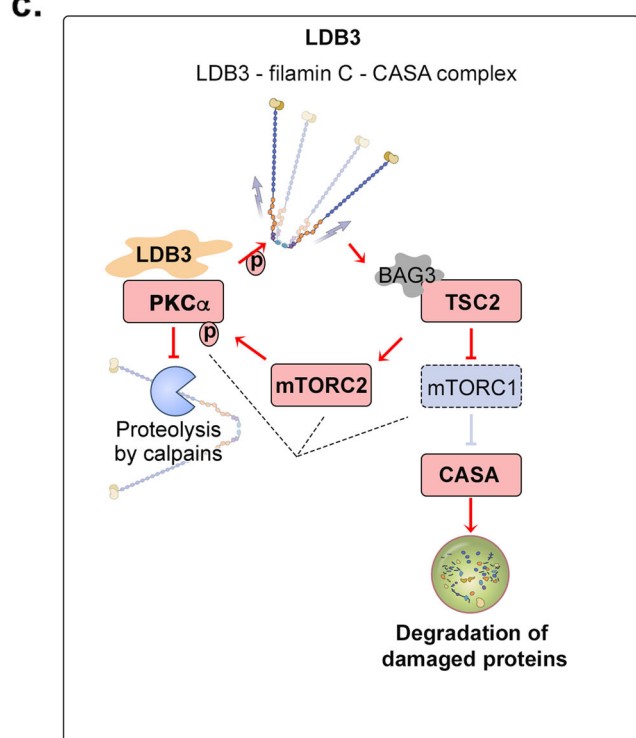

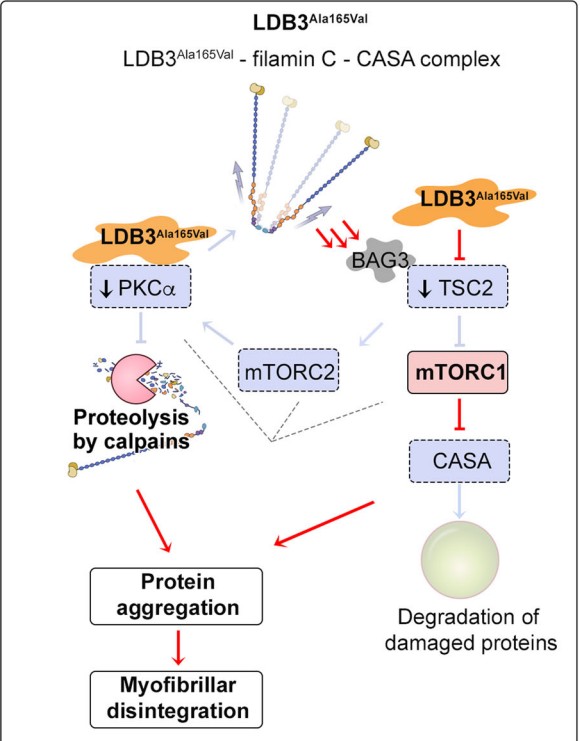

proteins[42,43]. Taken together, reduction in PKCα likely predisposes filamin C to calpain-mediated proteolysis leading to protein aggregation in skeletal muscle of $Ldb3^{Ala165Val/+}$ mice (Fig. 7c).

The aggregation of damaged filamin C aggravates the autophagy pathway in muscle fibers of $Ldb3^{Ala165Val/+}$ mice. This is evidenced by marked accumulations of the CASA chaperones in the muscle fibers containing protein aggregates and increased levels of the autophagy markers after colchicine blockage in skeletal muscle of 4-month-old mice. However, it appears that the strain on the cellular degradation system ultimately limits the capacity of the CASA chaperones to remove the damaged misfolded proteins and causes myopathy. This connection is supported by the hallmark MFM pathology changes in the $Ldb3^{Ala165Val/+}$ mice that resemble muscle pathology caused by mutations in filamin C and CASA components BAG3, DNAJB6,

and HSPB8, which have been proposed to negatively affect CASA function[23,44–48]. The CASA pathway constantly operates at the Z-disc and removes mechanically damaged proteins such as filamin C, thus it differs from the atrophy-driven autophagy pathways[49]. Recruitment of TSC1:TSC2 – mTORC1 assemblies by BAG3 to skeletal muscle Z-disc is required for the disposal of damaged filamin C through the CASA pathway[35]. Within the complex, TSC1 stabilizes TSC2[50,51], whereas TSC2 acts as a GTPase activating protein and integrates signals from various kinases leading to mTORC1 regulation in the cell[33]. It is possible that the reduction of TSC2 in skeletal muscle of $Ldb3^{Ala165Val/+}$ mice perturbs CASA function, leading to filamin C aggregation in muscle fibers (Fig. 7c). In addition, chronic changes in PKCα and TSC2-mTORC1 levels likely affect the balance of lysosomal autophagy in skeletal muscle of $Ldb3^{Ala165Val/+}$ mice[52].

**Fig. 7 Model of proposed mechanism for protein aggregation and Z-disc disassembly by LDB3 p.Ala165Val mutation. a, b** Diagrams show LDB3 – filamin C – CASA complex interactions at skeletal muscle Z-disc relevant to MFM pathogenesis. LDB3 binds to the mechanosensor Ig 17–21 repeats (orange) in filamin C where the binding sites for the MFM-associated proteins BAG3 and myotilin, as well as a myopathy-associated integrin complex have been identified[25,36,37]. LDB3 binds to myotilin and PKCα, which regulates filamin C stability through phosphorylation (P)[14–16,34]. LDB3 interacts with HSPA8, a BAG3 partner that together with the MFM-associated HSPB8 chaperone degrades damaged filamin C through CASA pathway (adapted from[25]). BAG3 cooperates with the HSPA8-associated ubiquitin ligase STUB1 and its partner UBE2D in the ubiquitination of chaperone-bound filamin C, which is recognized by sequestosome-1 leading to lysosomal disposal. These interactions are unaffected by the MFM-associated LDB3 p.Ala165Val mutation. **c** Model of LDB3 p.Ala165Val mutation effects on the integrity of LDB3 – filamin C – CASA complex interactome under mechanical strain. The activity of each protein and interaction is presented with color, blue for decrease and red for increase. An arrow indicates an activation, a bar at the end of an edge indicates an inhibitory interaction. A downward arrow within a box indicates decrease in the protein levels. Under normal conditions, LDB3 enables PKCα-mediated phosphorylation of filamin C and other interacting proteins at the Z-disc and protects these proteins from proteolysis by calpains. The CASA pathway constantly operates at the Z-disc mediating degradation of large cytoskeleton components including filamin C damaged during mechanical strain[49]. The CASA activity depends on local mTORC1 inhibition through BAG3-recruited TSC2:TSC1 signaling[35]. PKCα is modulated by TSC2-mTORC2 signaling and the kinase may regulate mTOR assemblies including spatial localization (dashed lines)[53–56]. The LDB3 p.Ala165Val downregulates PKCα and TSC2-mTOR, two signaling proteins monitoring the integrity of the Z-disc assembly. Decreased PKCα promotes proteolysis by calpains leading to aggregation of damaged proteins. Reduced TSC2-mTOR together with increased strain on capacity of degradation pathway leads to impaired CASA function aggravating damaged protein aggregation, eventually leading to the Z-disc disassembly and myofibrillar disruption.

In $Ldb3^{Ala165Val/+}$ muscle, the decrease in PKCα levels may be related to reduced TSC2 through mTORC2 inhibition[53,54]. PKCα is one of the best-characterized substrates of TSC2-mTORC2 that potentially regulates subcellular localization of mTORC1 and mTORC2 components, thereby permitting selective activation of specific targets[55,56]. It remains to be determined whether the MFM-associated mutation affects subcellular translocation of mTOR complexes to the Z-disc of skeletal muscle upon mechanical strain. The RPPA showed that the levels of TSC1 and Akt were unchanged in $Ldb3^{Ala165Val/+}$ mice, thus suggesting alternative mechanisms for TSC2 destabilization[50,51,57]. Furthermore, expression of total and phosphorylated S6K and 4E-BP1, two key regulators of translation was unaffected[57], suggesting that the MFM mutation does not affect the anabolic process of protein synthesis in skeletal muscle of $Ldb3^{Ala165Val/+}$ mice.

Patients with LDB3 p.Ala165Val mutation typically develop a late-onset myopathy[6,7,22]. Similarly, $Ldb3^{Ala165Val/+}$ mice developed normally until 3 months of age, and filamin C aggregation is not visible until 4 months of age. The mutant mice did not develop respiratory failure, cardiomyopathy, or neuropathy, which reflects the clinical disease in patients known to have the same mutation and is consistent with the fact that only negligible expression of exon 6 containing LDB3 isoforms is found in mouse heart and nervous system[9]. In contrast, BAG3 mutations are associated with severe congenital MFM in children, suggesting that BAG3 mutations dramatically impact CASA and other protein degradation mechanisms, leading to a severe widespread disease[46,47]. Selective vulnerability of skeletal muscle to the LDB3 p.Ala165Val mutation may be related to high expression of the mutated actin-binding domain in muscle tissue[17]. Late-onset disease in $Ldb3^{Ala165Val/+}$ mice and patients with the same mutation suggests that the CASA failure is limited to select substrates and that alternate protein quality control mechanisms may compensate to some extent until a later age and disease stage[47]. Moreover, age-induced increased dependence on the BAG3-mediated autophagy for disposal of damaged proteins that cannot be degraded by the proteasome may contribute to late-onset MFM[58].

In conclusion, the LDB3 p.Ala165Val mutation defies the prevailing thinking for pathogenesis of protein aggregation in neurodegenerative diseases[59]. The mutant protein does not seed the aggregation process through misfolding or self-aggregation[18], but triggers aggregation of filamin C and its chaperones at the Z-disc of skeletal muscle. Our findings indicate new roles for LDB3 as a modulator of mechanosensing through interactions with filamin C and its chaperone, as well as PKCα in the Z-disc

of skeletal muscle. And, they demonstrate that the LDB3 p.Ala165Val mutation leads to Z-disc disassembly and protein aggregation by affecting the dynamic signaling pathways mediated by PKCα and TSC2-mTOR at a major mechanosensory hub in skeletal muscle. Ultimately, our results support signaling modulation by small molecules or allele silencing as potential therapeutic strategies for this debilitating degenerative disease with as of yet no approved treatment.

## Methods

**Mice.** Wildtype mice (C57BL/6 N) were from Charles River laboratories (Wilmington, MA). The *β-actin-cre* mice were described previously[60]. Heterozygous LDB3-null mice (C57BL/6N-A$^{tm1Brd}$/a *Ldb3$^{tm2a(EUCOMM)Hmgu}$*/BcmMmucd) were from the Mutant Mouse Resource & Research Centers[61]. All animals were housed in the animal care unit of the NINDS according to the NIH animal care guidelines. All animal studies were authorized by the Institutional Animal Care and Use Committee of the NINDS.

**Antibodies.** Primary antibodies are listed in Supplementary Table 3. Antibodies against LDB3 were generated by immunizing rabbits with a peptide corresponding to amino acid residues 116–130 encoded by exon 6 of human LDB3 (NP_001073585; LDB3ex6ab; Supplementary Fig. 5a). We validated the LDB3ex6ab antibody for immunoblotting and immunofluorescence using transfected COS-7 cells and tissues of *Ldb3$^{+/+}$* and *Ldb3$^{-/-}$* mice (Supplementary Fig. 5b–d). Alexa Fluor-, IRDye-, and HRP- conjugated goat anti-mouse or anti-rabbit secondary antibodies were purchased from Invitrogen, LI-COR biosciences, and Jackson ImmunoResearch Laboratories, respectively.

**DNA constructs.** Primers are listed in Supplementary Table 4. A fragment encoding the rod domain 2 Ig17-21 of human filamin C (nucleotides 5629-7352; NM_001458.4) was amplified by PCR with filamin C cDNA clone (Ari57A02; RIKEN BioResource Research Center) as a template and cloned into pCMV-HA vector (Clontech). Full-length human HSPA8 cDNA (I.M.A.G.E. clone ID 2899894; NM_006597) was amplified by PCR and cloned into pCMV-HA vector. Constructs for FLAG-tagged human skeletal muscle WT and Ala165Val LDB3-LΔex10 (NM_001080114), GFP-tagged human skeletal muscle LDB3-S (NM_001080116), and pGBKT7-LDB3 exons 8-10-11 and 8-11Δex10 have been described previously[17].

**Generation of *Ldb3$^{Ala165Val/+}$* mice.** Strategy for generating the p.Ala165Val knock-in mutation in the mouse *Ldb3* gene is shown in Supplementary Fig. 1a. The RP23-244C16 Bacterial Artificial Chromosome (BAC) clone containing the mouse *Ldb3* gene (C57BL/6 N) was modified to reproduce the MFM causative p. Ala165Val mutation (chr14:34571772 C > T; GRCm38/mm10; C57BL/6 N), by a galK-based recombineering in SW102 cells, as described previously[62]. The targeting construct homologous to the region flanking the exon 6 mutation was generated by PCR using two 100-mer oligonucleotides with 20 bp overlap at the 3′ end. Next, the *Ldb3* genomic region between introns 3 and 8 (chr14:34567235–34578097) containing the mutation was retrieved from the modified BAC into PL253 gene-targeting vector by "gap repair", as described previously[63]. And finally, a neomycin selection marker cassette (PGK-neo), flanked by loxP sites, was inserted within intron 6 between chr14:34570741 and 34570742 base pairs in the direction of gene transcription by gap repair using PL452 vector containing ~500 bp homology arms

flanking the insertion site, as described previously[63]. The final modified PL253 gene-targeting vector was linearized with *Not I* and electroporated into embryonic stem (ES) cells generated from B6x129 hybrid mice. Targeted ES cells were identified by Southern blotting and confirmed by Sanger sequencing. Positive ES cell clones were injected into blastocysts to obtain chimeric mice as described previously[64]. The presence of the selection cassette disrupted allelic expression in *Ldb3*$^{flox-Ala165Val/+}$ mice. Functional knock-in lines were activated by crossing *Ldb3*$^{floxed-Ala165Val/+}$ mice with β-actin-cre transgenic mice (C57BL/6 N). Offsprings were bred with wildtype mice to remove Cre-recombinase. The resulting *Ldb3*$^{Ala165Val/+}$ mice were backcrossed for at least 5 generations on C57BL/6 N background.

**Targeted gene sequencing of the *Ldb3* recombined region**. Mouse genomic DNA was PCR amplified using primer sets described in Supplementary Table 4. DNA library was prepared using the Nextera library preparation kit (Illumina). The concentrations of the indexed libraries were analyzed on the Agilent 2200 TapeStation using the D1000 Kit (Agilent Technologies). Equimolar amounts of the indexed libraries were pooled to obtain a 4 nM library mixture. After denaturing and further diluting, the final 12 pM library was loaded into an Illumina cartridge. Sequencing was performed using the Illumina MiSeq Reagent Kit v2 (500 Cycles) on the Illumina MiSeq instrument following the manufacturer's instructions.

**Southern blotting**. Tail DNA was extracted by digestion in buffer (1 M Tris-HCl pH 8.5, 0.5 M EDTA, 5 M NaCl, 10%SDS, 10 mg/ml proteinase K) overnight at 55 °C with continuous shaking in low speed. Lysates were cleared by centrifugation at 14,000 rpm for 5 min, DNA is precipitated in equal volume of isopropanol, pelleted at 14,000 rpm for 5 min, washed three times in 70% ethanol, air dried, and rehydrated in water. Digoxigenin-labeled DNA probes hybridized to the 5′ (nucleotides 34579195–34579596; NC_000080.6) and 3′ (nucleotides 34564772–34565187) flanking sequence of the recombined *Ldb3* genomic region were used. Mouse genomic DNA samples (10 μg) were digested with *EcoRV* (New England Biolabs) overnight at 37 °C, separated on a 0.7% agarose gel over 20 h and blotted onto a positively charged nylon membrane (Roche Diagnostic) by capillary transfer for 48 h. Hybridization was done overnight at 45 °C with agitation at 10 rpm. Membranes were washed twice with 2X SSC containing 0.1% SDS for 5 min at 65 °C and then twice with 0.5X SSC containing 0.1% SDS for 15 min at 65 °C. DIG-labeled probes after hybridization to target DNA were detected using DIG High Prime Detection kit II (Roche Diagnostic). Images were acquired on a ChemiDoc imager (BioRad) using Image Lab software (version 5.2).

**Mouse genotyping**. Mouse genotypes were identified by PCR using primers to detect the residual loxP site as well as a SNP genotyping assay on genomic DNA (Supplementary Fig. 1d, e). The presence of mutation was validated at gene and transcript levels by Sanger sequencing. Primers and probes are listed in Supplementary Table 4. Genomic DNA was extracted from mouse tails or ear punches using Direct PCR reagent (Viagen Biotech, Los Angeles, CA). Genotyping PCR reactions were done using Taq DNA polymerase (GE Biosciences) and primers flanking the residual loxP site in intron 6 of the mutant allele for *Ldb3*$^{Ala165Val/+}$ mice. TaqMan SNP genotyping assays were run on QuantStudio 6 Flex Real-Time PCR System (Applied Biosystems) using primers flanking the mutation and Taq-Man allele-specific probes (Life Technologies). For confirmation by Sanger sequencing, DNA fragments were amplified using Platinum Taq DNA polymerase High Fidelity (Thermo Fischer Scientific) and primers flanking the mutation in *Ldb3* exon 6.

**RNA Isolation from tissue**. Mouse skeletal muscle tissue was homogenized using Ambion Trizol Reagent (Life Technologies) and RNAse-free 0.5 mm zirconium oxide beads in a Bullet Blender homogenizer (Next Advance, Troy, NY). Total cellular RNA was isolated by phase separation using 1-Bromo-3-Chloropropane followed by purification with RNAeasy Mini Kit (Qiagen). RNA integrity was assessed using Agilent 2100 Bioanalyzer. Samples with RNA integrity number >8.5 were used for all assays. RNA was stored at −80 °C.

**Reverse transcription (RT)**. RT was done in 20 μl volume using 1μg total cellular RNA and Superscript VILO master mix or High Capacity cDNA RT kit containing random hexamers according to manufacturer's recommendations (Thermo Fischer Scientific). The cDNA samples were stored at −20 °C.

**Real-time quantitative PCR (qPCR)**. qPCR assays were run on QuantStudio 6 Flex Real-Time PCR System (Applied Biosystems, Life Technologies) using Taq-Man probe for *Ldb3* spanning constitutively spliced exons 2 and 3 (NM_001039071.2; Mm01208763_m1), thus allowing assessment of total *Ldb3* transcript levels. The ΔΔC$_T$ method was used for quantification of *Ldb3* transcript expression relative to a reference gene (Supplementary Fig. 1c). Expression levels were normalized to *Myom1* and *Rer1* genes (Taqman; Mm00440394_m1 and Mm00471276_m1, respectively) because these genes showed the lowest intragroup and between group variability for wildtype and mutant mice in RNAseq and

microarray data ($n = 20$ mice). All primers and probes were validated for optimal amplification efficiency and all samples were used within the dynamic range of the assays. Linearity was established over four orders of magnitude using cDNA titration series. Multiple biological and technical replicates were used in each assay ($n \geq 3$). At least three independent RT-qPCR assays were performed for each sample.

**Gross motor performance and muscle strength testing**. The rotarod test was done as described previously (RotaRod Series 8, IITC Life Sciences, CA, USA)[65]. Four paw grip strength was assessed by BIOSEB grip strength meter (BIOSEB, Vitrolles, France) using a method adapted from Treat-NMD Standard Operating Procedure (DMD_M.2.2.001). Five consecutive measurements of peak pull force were performed per day at 2 min intervals. The maximum strength normalized to body weight was used for analysis. These test sessions were performed during the afternoon hours of the light cycle (11 AM to 2 PM) in the behavioral core facility. A two-way analysis of variance with post hoc Bonferroni correction for multiple comparisons was used to determine effects for age and genotype (GraphPad Prism 8). The four-limb hang test was adapted from Treat-NMD Standard Operating Procedure (DMD_M.2.1.005). Mice were allowed to grasp the wire grid by all four limbs before inverting at 40 cm above a layer of bedding (NIH Instrumentation Core Apparatus). The hang time (s) was measured from the time the grid was inverted to the time that the mouse fell off the wire grid using a stopwatch. The test was repeated in mice two more times with 5 m of rest period in between testing. The longest suspension time from the three trials was used for analysis. The mouse body weight (g) was measured immediately before testing. The holding impulse [N*s; weight (g) × 0.00980665 (N/g) × the maximum hang time (s)] was compared between the groups. These tests were done during the afternoon hours of the light cycle (3 PM to 5 PM) in the behavioral core facility. An unpaired *t*-test (two-tailed) was applied for group comparison (GraphPad Prism 8). We used alpha level of 0.05 for all statistical tests. Investigators were blinded to the genotype of the mice. See Supplementary Data 2 and 4 for source data.

**In vitro muscle contractility measurements**. Mice were anesthetized by intraperitoneal tribromoethanol (250 mg/kg). Tissue isolation and measurement were adapted from Treat-NMD Standard Operating Procedure (DMD_M.1.2.002) with some modifications. Stainless steel hooks were fastened onto the proximal and distal tendons of the extensor digitorum longus muscle using 4.0 silk suture before muscle removal. The muscle was quickly installed into the measurement chamber containing Ringer's solution (1.2 mM NaH$_2$PO$_4$, 1 mM MgSO$_4$, 4.83 mM KCl,137 mM NaCl, 2 mM CaCl$_2$, 10 mM glucose, 24 mM NaHCO$_3$, pH 7.4) pre-warmed to 37 °C and equilibrated with 95% O$_2$/5% CO$_2$. An Aurora 1200 A in vitro dual lever system (Aurora Scientific, Aurora, ON, Canada) was used for all recordings. For each muscle, optimal length (L$_o$) and supramaximal (110% of maximal) current were determined at the beginning of each session using a series of isometric stimulations (1 Hz, 0.2 ms pulse width) separated by 20 s. The measured Lo for isometric force measurements was similar between mutant and wildtype mice (mean ± SEM: 11.3 mm ± 0.2 mm versus 11.6 ± 0.4 mm, $p = 0.28$). A force-frequency curve was established using a series of tetanic isometric stimulations (20–250 Hz, 0.2 ms pulse width, 500 ms train duration). The muscle cross-sectional area (CSA) was calculated using the formula: muscle mass (mg)/[L$_o$ (mm) × 0.44 × 1.06 mg/mm$^3$]. Data are presented as specific isometric force (mN/mm$^2$), calculated using the formula: Force (mN)/CSA (mm$^2$). A two-way analysis of variance with post hoc Bonferroni correction for multiple comparisons was used to determine effects for genotype (GraphPad Prism 8). Investigators were blinded to the genotype of the mice. See Supplementary Data 3 for source data.

**Histology**. Frozen 10 μm thick transverse muscle sections were stained with modified Gomori trichrome and nicotinamide adenine dinucleotide dehydrogenase-tetrazolium reductase (NADH-TR) enzymatic activity using standard procedures. Heart tissues obtained from mice were fixed in 4% paraformaldehyde (PFA) overnight at 4 °C and then cryoprotected in 30% sucrose at 4 °C. For histological examinations, hearts were embedded into wax blocks of paraffin and 5 μm thick sections were stained with hematoxylin and eosin.

**Immunofluorescence**. Frozen 10 μm thick transverse muscle sections were fixed in 4% PFA for 10 min, permeabilized and blocked with 2.5% goat serum, 0.2% Triton X-100/phosphate buffered saline (PBS) for 1 h. Subsequently, slides were incubated with appropriate primary antibody (Supplementary Table 3) diluted in 1.25% goat serum with 0.1% Triton X-100/PBS overnight at 4 °C. Sections were incubated with fluorophore-conjugated secondary goat anti-mouse or anti-rabbit antibodies (Jackson ImmunoResearch Laboratories) for 1 h at room temperature before being mounted and sealed. Images were acquired on a Leica confocal microscope (TCS SP5 576 II) equipped with a 40×/NA 1.3 oil immersion Plan-Apochromat objective lens and processed with Adobe Photoshop Creative Cloud v2017. Identical settings were used on all images of a given experiment and comparisons were made for muscle sections stained on the same slide.

For longitudinal muscle sections, mice were transcardially perfused with 4% PFA in PBS and skeletal muscle tissues were post-fixed in same buffer overnight at 4 °C. Muscle tissues were cryoprotected in 30% sucrose for 48 h at 4 °C, embedded in Optimal Cutting Temperature compound, and stored at −80 °C until use.

Longitudinal sections (8 µm) were collected onto slides and antigen retrieval was carried out by boiling slides in citrate buffer (pH 6.0) for 10 min before immunostaining. Images were acquired on a Leica confocal microscope (TCS SP5 576 II) using a 63×/NA 1.4 oil immersion Plan-Apochromat objective and processed as described for frozen sections.

For fiber-type staining, unfixed frozen transverse soleus and vastus muscle sections (10 and 8 µm, respectively) were blocked with 10% goat serum/PBS for 1 h and then incubated overnight at 4 °C in antibodies to myosin heavy chain (MyHC) types I, IIA, IIB, and IIX (BA-F8, SC-71, BF-F3, and 6H1, respectively, Developmental Studies Hybridoma Bank; Supplementary Table 3). Fluorescence-conjugated secondary antibodies to different mouse immunoglobulin subtypes (IgG2b – Alexa Fluor 488 or 568, and IgM – Alexa Fluor 568, ThermoFisher Scientific; CY5-IgG1, Abcam) were applied for 1 h at room temperature to visualize MyHC expression. In addition, wheat germ agglutinin – Alexa Fluor 488 conjugate (WGA; ThermoFisher Scientific) was applied to stain muscle membrane for fiber type–size quantitation. Sections were mounted with Fluoromount – G with DAPI (Southern Biotech). Images were obtained using a Leica epifluorescent microscope (DMI6000 SD) equipped with a motorized stage. Images were acquired in Leica Application Suite (LAS) X software with a 20X/NA 0.4 objective. The LAS X Navigator tool was used to take images from adjacent fields and digitally stitch them (with 10% overlap) to form a single image of the entire soleus muscle cross-section for analysis. For vastus muscle, nonoverlapping areas were photographed for analysis. Myonuclei counting, muscle fibers with sarcoplasmic protein aggregate quantitation, muscle fiber typing (using the counting tool), and minimal Feret's diameter measurement for fiber size were performed for all fibers excluding those on the edges by two individuals using open-source image processing software Image J/Fiji (National Institutes of Health, Bethesda, MD).

**Electron microscopy**. Mice were transcardially perfused with 2% glutaraldehyde and 2% paraformaldehyde in PBS and skeletal muscles were post-fixed in same buffer in 0.1 M sodium cacodylate. The tissue embedding and staining were carried out in the NINDS electron microscopy facility using a standard approach. Images were acquired on an electron microscope (JEOL 200CX, Jeol, Inc.) and processed with Adobe Photoshop Creative Cloud v2017.

**Immunoblotting**. Protein was extracted from transverse muscle midbelly cryosections ($n = 15$, 10 µm thick) using SDS buffer (100 mM Tris-HCl; pH 6.8, 4% SDS, 20% glycerol, 2% β-ME, 25 mM EDTA pH 8) supplemented with protease inhibitor (Roche). Equal volumes were loaded for relative protein quantification by immunoblotting. For PKCα and TSC2 levels, protein was extracted from muscle tissue (30 mg) homogenized in 600 µl T-PER Reagent (Thermo-Fisher) supplemented with protease and phosphatase inhibitors (Roche) using polytron homogenizer (ThermoScientific). Supernatants were collected after centrifuging the samples at $10,000 \times g$ for 5 min at 4 °C. Proteins were quantified using Bradford assay. Proteins resolved on Tris-Glycine polyacrylamide gels were transferred to nitrocellulose membrane (Life Technologies). After blocking in 5% BSA, 0.2% Tween 20/PBS for 1 h at room temperature, the membranes were incubated with primary antibodies in 5% BSA, 0.1% Tween 20/PBS overnight at 4 °C and then with secondary antibodies for 1 h at room temperature. Secondary antibodies were conjugated with IRDye (LI-COR Biosciences). Odyssey Imaging Systems and Image Studio Lite software were used for detection and quantification, respectively (LI-COR Biosciences). HRP-conjugated secondary antibodies (Jackson ImmunoResearch Laboratories) and the ChemiDoc XRS + Molecular Imager (Bio-Rad, Hercules, CA) were used for detection in GST pulldown assay and for blots shown in Supplementary Fig. 5. At least three independent assays were performed for each sample. Multiple biological replicates were used in each assay ($n \geq 3$). See Supplementary Fig. 6 for full-length blot images and Supplementary Data 1, 8, and 9 for source data.

**Immunoprecipitation**. Mouse tibialis anterior muscle lysates were homogenized on ice using polytron homogenizer in buffer (20 mM Tris-HCl pH 7.5, 100 mM KCl, 5 mM EGTA, 1% Triton X-100, and EDTA-free protease inhibitor). The homogenized lysates were incubated for 1 h at 4 °C with gentle agitation and then centrifuged at 3000 g for 30 min at 4 °C. The pellet containing myofibrillar fraction was homogenized one more time and washed twice in buffer (20 mM Tris-HCl pH 7.5, 100 mM KCl, 1 mM DDT, and EDTA-free protease inhibitor) before resuspending in storage buffer (20 mM Tris-HCl pH 7.5, 100 mM KCl, 1 mM DDT, 20% glycerol). The lysates containing myofibrillar fraction were precleared with protein A Dynabeads (Invitrogen) for 3 h at 4 °C. The cleared lysates were incubated with either mouse anti-LDB3 antibody (Abnova) or mouse IGg2b (Cell signaling) for overnight at 4 °C and subsequently with protein A Dynabeads for 3 h at 4 °C. The beads retaining immune complexes were washed five times with wash buffer (50 mM Tris-HCl pH 8.0, 0.1% Triton X-100, 5 mM EDTA) and then heated in SDS sample loading buffer for 5 m at 95 °C. The eluates and lysates were analyzed by immunoblotting. Immunoprecipitation of tagged proteins from transfected COS7 cell lysates was done as described previously[17].

**GST pulldown assay**. The pulldown assays using recombinant GST-tagged LDB3 proteins were done as described previously[17]. At least three independent experiments were performed using the vastus muscle lysates of three wildtype mice.

**Autophagosome flux**. Measurement of autophagosome flux in mouse skeletal muscle tissue using lysosomal blockade by colchicine was done as described previously[21]. Mice were treated with intraperitoneal colchicine (0.4 mg/kg/day; Sigma) or vehicle alone (PBS pH 7.4) for 3 days. The day following the last injection, hindlimb muscles were collected for protein isolation. Levels of autophagosome markers LC3-II and sequestosome 1 (p62) were quantified by immunoblotting. GAPDH was used as a loading control. The samples were run on two gels (one mini gel could not accommodate all sample loadings) in the same electrophoresis tank. The protein transfer from both gels was done simultaneously on a single membrane which was then processed for immunoblotting. For densitometric analysis, all the band intensities were normalized to that of one PBS-treated $Ldb3^{+/+}$ sample in each membrane. See Supplementary Fig. 6b for full-length blot images. Autophagosome flux values were derived by subtracting corresponding protein values in vehicle treated from the colchicine treated conditions ($Ldb3^{+/+}$ COL minus $Ldb3^{+/+}$ VEH and $Ldb3^{Ala165Val/+}$ COL minus $Ldb3^{Ala165Val/+}$ VEH). Three independent assays were performed for each sample. See Supplementary Data 5 for source data.

**Yeast two-hybrid screen**. The yeast two-hybrid (Y2H) screen was performed by Hybrigenics Services (Paris, France) using human $LDB3$ exons 8 and 11 corresponding to LDB3-LΔex10 as a bait (amino acid 221 – 320; NM_001080114; NP_001073583), and a human skeletal muscle cDNA library for interactors.

**Yeast two-hybrid pairwise assay**. Pairwise Y2H assays were done using the Matchmaker Gold system (Clontech), as described previously[17].

**RPPA analysis**. The RPPA analysis on proteins isolated from frozen vastus muscle of 4- and 8-month-old $Ldb3^{+/+}$ and $Ldb3^{Ala165Val/+}$ mice ($n = 5$ per group, total 20 mice) was performed by the Functional Proteomics RPPA Core facility at MD Anderson Cancer Center (Houston, TX). Tissue lysate samples were serially diluted two-fold for 5 dilutions (undiluted, 1:2, 1:4, 1:8; 1:16) and arrayed on nitrocellulose-coated slides in an $11 \times 11$ format to produce sample spots. Sample spots were then probed with 248 antibodies (159 validated for RPPA) against total and phosphorylated proteins by a tyramide-based signal amplification approach and visualized by DAB colorimetric reaction to produce stained slides. Stained slides were scanned on a Huron TissueScope scanner (Huron Digital Pathology, Ontario, Canada) to produce 16-bit tiff images. Sample spots in tiff images were identified and their densities quantified by Array-Pro Analyzer 6.3. Relative protein levels for each sample were determined by interpolating each dilution curve produced from the densities of the 5-dilution sample spots using a "standard curve" (SuperCurve) for each slide (antibody). SuperCurve 1.5.0 (via Super-CurveGUI_2.1.1) was constructed by a script in R-package. All relative protein level data points were normalized for protein loading and transformed to linear values, which were designated "Normalized Linear" or "NormLinear" and were used for expression analysis. Only antibodies with quality control scores above 0.8 in the heatmaps were included in analysis. To identify differentially expressed proteins between age and gender-matched $Ldb3^{Ala165Val/+}$ mice and $Ldb3^{+/+}$ littermates, raw abundances detected per sample were log transformed (base 2), median centered, and variance-stabilized (median absolute deviation) using a R-package. Proteins having an absolute difference of means $\geq 1.5 \times$ and a corrected $p \leq 0.05$ (Welch modified $t$-test under Benjamini–Hochberg false discovery rate multiple comparison correction; rounded to the nearest hundredth) were deemed to be differentially expressed between the groups. See Supplementary Data 6 and 7 for source data.

**Statistics and reproducibility**. The statistical analyses were performed using either GraphPad Prism 8 or R package. All individual data points are plotted as dot plot or bar-dot plot and presented as mean ± SEM or SD throughout the manuscript. Statistical testing method, sample size, replicates of experiments, and the $p$ value are indicated in figure legends. A two-way ANOVA using the Bonferroni's multiple comparison test or an unpaired two-tailed $t$-test were done to determine differences between groups. Statistical significance was defined as $p < 0.05$. Welch modified $t$-test under Benjamini–Hochberg false discovery rate multiple comparison correction was applied to identify differentially expressed proteins between genotypes in the RPPA analysis using a R-package. Statistical significance was defined as absolute difference of means $\geq 1.5 \times$ and a corrected $p \leq 0.05$.

**Reporting summary**. Further information on research design is available in the Nature Research Reporting Summary linked to this article.

## Data availability

The datasets generated during and/or analyzed during the current study are available from the corresponding author on reasonable request. Source data underlying plots shown in main figures are provided in Supplementary Data. Full blots are shown in Supplementary Fig. 6 in Supplementary Information.

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

## Acknowledgements

The study was supported by the Intramural Research Program (IRP) of the National Institute of Neurological Disorders and Stroke (NINDS) and National Cancer Institute (NCI). Mouse behavior studies were supported by the National Institute of Mental Health IRP Rodent Behavioral Core (MH002952). The authors thank Eileen Southon, Susan Reid, and Lino Tessarollo (NCI) for knock-in ES cell production and microinjections; Virginia Tanner-Crocker and Dr. Susan Cheng for tissue embedding and sectioning for electron microscopy (NINDS Electron microscopy Core), Dr. Michael Eckhaus for heart histology (Division of Veterinary Resources, Diagnostic and Research Services Branch), Nathan Sarkar (NINDS) and Kory Johnson (NINDS Bioinformatics Section) for bioinformatic support, and Alan Hoofring for designing diagram figures (NIH Medical Arts Design Section). The Functional Proteomics Reverse Phase Protein Array Core Facility at MD Anderson Cancer Center is supported by NCI #CA16672 grant.

## Author contributions

A.M. designed the experiments, obtained funding, and supervised the study. P.P., Y.B., K.S., J.M., C.O., R.O., I.M., M.K., J.H., S.S., S.K.S., and A.M. acquired, analyzed, and interpreted the experimental data. P.P., Y.B., K.S., and A.M. prepared the figures. P.P., Y.B., and A.M. wrote the manuscript. P.P and Y.B. contributed equally. All authors approved the content of the manuscript.

## Funding

## Competing interests

The authors declare no competing interests.
