## [Peer Review File · Communications Biology]

Reviewers' comments:

Reviewer #1 (Remarks to the Author):

This paper examines a mouse model of LDB3 myofibrillar. It demonstrates the appearance of pathology consistent with myofibrillar myopathy and demonstrates the filamin C aggregation prior to aggregation of LDB3, suggesting it is a functional deficit of LDB3 rather than a conformational change that triggers protein aggregation. The authors also identify a reduction in PKC α and TCS2 in the muscle suggesting a potential disease mechanism.

The summary diagrams provided are a useful guide to the reader. The authors have made a very useful model of LDB3 MFM and there are some interesting observations.

There are, however, many serious concerns that need to be addressed. Several conclusions are made without supporting evidence, and for others it would appear that the interpretation provided is not correct, or limited data is over interpreted.

The specific issues identified are detailed below.

- 1) Line 51 The authors report that based on the Mayo Clinic cohort LDB3 A165V is the cause of the majority of MFMs. However, in the reference related to this claim (reference 7) it states that 11 of the 54 patients had mutations in ZASP with A165V in only 3 of them. Therefore, the majority of MFMs are not due to LDB3 mutations, and even within the Mayo Clinic cohort LDB3 cases, A165V is not the most common mutation. This statement needs to be corrected.
- 2) Figure 1e – it is not clear what comparison is being highlighted. It would appear that it is highlighting a significant difference in force between WT mice at 80 and 180Hz. This should be corrected to make clear what is being shown. The legend also indicates a t-test was used and yet there are many more than 2 groups, and therefore the test applied is not appropriate.
- 3) A serious issue that first appears in figure 2 but is present for all of the pathology is the complete absence of control images for immunolabelling, histology, and electron microscopy. Control images, of WT littermates must be included for every image shown in figure 2 and subsequent figures.
- 4) In Figure 2g the statistical comparison is showing the effect of colchicine treatment and should be comparing flux between the WT and mutant mice.
- 5) Figure 2g – in addition to the graph not showing the correct comparison the interpretation does not appear to be correct. The authors conclude the autophagy is dysfunctional, but flux is higher in the mutant mice, instead suggesting that autophagy is increased in the mutant animals. The interpretation of the data needs to be re-examined, and the conclusions altered if necessary.
- 6) Line 153 'Data not shown'. The data should be provided to support all statements in the manuscript. This includes line 168 where it is reported that soleus and vastus were examined, but only the vastus is shown.
- 7) Line 184 refers to disintegration occurring in mice at 6 months and older but no disintegration is shown and this, and subsequent statements of 'disintegration' need to be rephrased. The timepoint in the text (6 months) does not match the figure (4 and 8 months). The correct age of the mice needs to be checked and the text and figures should match.
- 8) Line 187. It is reported that CASA component aggregates were not observed in adjoining fibres but the adjoining fibres are not shown and should be included in the images.
- 9) Line 239 it is reported that CASA functions are impaired, but the function has not been examined and therefore analysis of the turnover of CASA targets needs to be included or the statement rephrased or removed.

- 10) Similarly line 240 refers to muscle fibre degeneration but this is not shown in any of the data provided.
- 11) Line 257 The authors report a change in protein levels for PKCalpha and TSC2. However, they conclude that this is responsible for the MFM phenotype. There is no evidence provided to support this conclusion.
- 12) Line 274 it is also concluded that disintegration occurs in fibers with aggregates, but disintegration is not shown. From the data provided no disintegration is apparant and the manuscript needs to be rephrased accordingly.
- 13) Line 321. It is again concluded that PKCalpha exquisitely controls Z-disc protein homeostasis. No examination of Z-disk protein turnover or levels is shown and the only examination of PKCalpha is the level of expression, with no functional analyses. Therefore, this conclusion is also not supported by evidence in the manuscript.

Minor issues:

- 1) Line 77 – ‘murine models of MFM have not yet been reported’ – LDB3 needs to be added to this statement as other murine models of MFM have been reported.
- 2) Figure 1 – it would be useful if the legend for the colours was included in the image. The brackets should align with the centre of the groups being compared.
- 3) The authors should check the approved nomenclature eg HSPB8 not Hsp22 and use throughout.

Reviewer #2 (Remarks to the Author):

The manuscript describes the analysis of a novel mouse knock-in model for the p.A165V mutation in the PDZ-LIM protein LDB3/Cypher/ZASP associated with human myopathy. The mouse model developed myofibrillar myopathy, characterized by muscle weakness and protein aggregation, mimicking the phenotype of human patients carrying the mutation. The authors showed aggregation of filamin C and its chaperones prior to the aggregation of mutant LDB3 in mutant mice, eventually resulting in myofibrillar disintegration. LDB3 was shown to directly bind to filamin C and its chaperone Hsc70, which, however, was not affected by the p.A165V mutation. PKC α and the negative mTOR regulator TSC2 were downregulated in the muscle of mutant mice, providing a likely explanation for the observed aggregation of damaged filamin C in mutant mice as PKC α and TSC2 are involved in regulation of filamin C stability and degradation of damaged filamin C, respectively. The manuscript is clearly written and convincingly elucidates a novel pathway for how LDB3 mutations can lead to myofibrillar myopathy, which may be of possible relevance also for other disease genes. The manuscript is thus of interest for others in the field and warrants publication. However, several issues need to be addressed as detailed below.

1. On page 6, line 110, it is stated that Ldb3A165V/+ mice showed a progressive decline in the grip strength test. However, this is only at 6 months vs. 3 months, while the difference between WT and mutant mice is less at 9 months. It would therefore be preferable to avoid the word “progressive”. Since the mice are compared at different time points, the statistical test should be done with ANOVA with appropriate post hoc analysis and not the unpaired t-test. Also in Figure 1e, ANOVA would be a more appropriate statistical test.

2. For the histological analyses in Figure 2, please state how many mice were analyzed per time

point? Also, maybe this information would be more appropriate in the figure legend than in the text (page 6, line 126-127). In Figure 2, pictures of the histological, immunostaining, and ultrastructural analyses are shown only for the mutant mice. Please show corresponding images from WT mice for comparison. The same applies for Figure 3 and 4.

3. In the immunofluorescence analyses in Figure 2c-d, would it be possible to quantify the percentage of cells showing sarcoplasmic accumulation of Z-disc proteins and ubiquitin in mutant mice compared to WT?

4. On page 7, line 138, it is stated that “Non-specific myopathic changes such as variability in fiber size, increases in internal nuclei, rounding of fibers, and increased in interstitial space were also observed”. However, no data are shown. The variability in fiber size could be due to altered fiber type distribution and/or fiber type specific fiber size. Therefore please provide quantification of fiber size and fiber type distribution, including fiber type-specific fiber size. It would also be useful to show images of the other mentioned myopathic changes, when possible with quantification, at least in the supplemental file.

5. In figure 2g, please show the individual data points on the graphs with the densitometric analysis. From the figure it appears as if the samples from WT and mutant mice were run on separate gels, although the main point of the analysis should be to compare protein levels in WT and mutant mice. Direct comparison of protein levels in WT vs. mutant mice treated or not with colchicine should be performed by ANOVA by running the samples from WT and mutant mice on the same gel. For LC3, please quantify the LCII/LC3I ratio, which is a more appropriate measure for the autophagic state

6. In Figure 3a, please show the WT control and quantify the amount of fibers with filamin C accumulation in mutant mice compared to WT. Please provide the n for both mutant and WT mice analyzed

7. In Figure 6, based on the trend towards reduced TSC2 levels at 8 months of age in the RPPA analysis, it would be preferable to show Western blot analysis for TSC2 also at 8 months of age.

8. It is stated that gender-matched littermates were used for the experiments but not whether preferably males or females were used for the analyses. Were the analyses done on mixed females and males? If so, were there any differences in for example grip strength and isometric force between genders and were the differences between WT and mutant mice similar in males and females?

Minor issues:

9. Page 8, line 163: “Patients usually present in advanced stages of myopathic disease” should be corrected to “Patients usually present with...”

10. Page 8, line 168: It is unclear what is meant with the statement: “The filamin C antibody staining was more strongly positive compared with LDB3 antibody in the vastus lateralis muscle fibers of patients sharing the same mutation”. The intensity of the two different antibodies cannot be directly compared. Please modify.

11. In Figure 5a, please write the number of replicates and how many times the assay was repeated.

Reviewer #3 (Remarks to the Author):

Pathak and Blech-Hermoni et al. present a very interesting functional study on the role of LDB3 in maintaining integrity of the Z-disc structure and therefore of the muscle myofibrils. The research on mutated protein is important for understanding one of the pathomechanisms of the myofibrillar myopathy. The manuscript is definitely of interest to the field, and only minor bits of information are missing.

The variant nomenclature should follow HGVS recommendations (<https://varnomen.hgvs.org/recommendations/>), especially:

- Protein descriptions should consistently use three-letter amino acid code
- Predicted consequences, when only DNA sequence (not RNA or protein sequence) was analysed should be given in parentheses, e.g. NP_001073583.1:(p.Ala165Val).
- Genetic variants should be at least once described in relation to an accepted reference sequence, e.g. NM_001080114.2:c.494C>T, NP_001073583.1:(p.Ala165Val). Please provide accession numbers of the transcripts and protein sequences.

The HGVS recommendations apply also to mouse variant.

It is worth mentioning that LDB3 p.Ala165Val mutation is likely benign according to ACMG criteria. <https://varsome.com/variant/hg38/rs121908334> Obviously phenotype and epidemiological data clearly indicates that the variant is in fact disease-causing.

Please note that several murine models of MFM were already reported. (Batonnet-Pichon et al. Myofibrillar Myopathies: New Perspectives from Animal Models to Potential Therapeutic Approaches. Journal of neuromuscular diseases. 2017)

We appreciate the critical evaluation of our work described in the manuscript, “Myopathy associated LDB3 mutation causes Z-disc disassembly and protein aggregation through PKCa and TSC2-mTOR downregulation”. We find the comments very constructive and helpful. We have revised the manuscript integrating the reviewers’ suggestions and believe that the result is a much stronger paper. We refer to line number below as per manuscript with track changes – no markup version.

Reviewer #1:

1. *Line 51 The authors report that based on the Mayo Clinic cohort LDB3 A165V is the cause of the majority of MFMs. However, in the reference related to this claim (reference 7) it states that 11 of the 54 patients had mutations in ZASP with A165V in only 3 of them. Therefore, the majority of MFMs are not due to LDB3 mutations, and even within the Mayo Clinic cohort LDB3 cases, A165V is not the most common mutation. This statement needs to be corrected.”*

Response 1: We modified the sentence in Introduction section (page 3, **lines 50 – 53**) as, “The dominant p.Ala165Val mutation in exon 6 of *LIM domain-binding 3* gene (*LDB3*; HGNC 15710; rs121908334 NM_001080114.2:c.494C>T, NP_001073583.1:(p.Ala165Val)) has been reported in several unrelated families of European ancestry^{6,7}.”

2. *Figure 1e – it is not clear what comparison is being highlighted. It would appear that it is highlighting a significant different in force between WT mice at 80 and 180Hz. This should be corrected to make clear what is being shown. The legend also indicates a t-test was used and yet there are many more than 2 groups, and therefore the test applied is not appropriate.”*

Response 2:

We used a two-way ANOVA multiple comparison test with post hoc Bonferroni correction (Test table and figure 1e below). We have also modified the manuscript text and legend accordingly as below.

- 2I. The Bonferroni’s multiple comparison test data table:

Bonferroni's multiple comparisons test	Mean Diff.	95.00% CI of diff.	Significant ?	Summary	Adjusted P Value
WT – Mutant					
20 Hz	12.97	-37.62 to 63.55	No	ns	>0.9999
40 Hz	17.91	-24.86 to 60.68	No	ns	0.6281
60 Hz	28.95	-43.35 to 101.2	No	ns	0.8008
80 Hz	34.67	-12.46 to 81.80	No	ns	0.1619
100 Hz	37.67	-3.161 to 78.49	No	ns	0.0696
120 Hz	39.27	3.851 to 74.70	Yes	*	0.0322

140 Hz	40.07	7.239 to 72.91	Yes	*	0.0214
160 Hz	39.60	9.840 to 69.36	Yes	*	0.0152
180 Hz	36.89	8.226 to 65.56	Yes	*	0.0191

2II. Revised Fig. 1 see panel e: the two-way ANOVA Bonferroni's significant p values shown.

2III. Fig. 1e legend, **lines 978 – 983**, Dot plot shows maximal isometric force (mN/mm²) generated by the extensor digitorum longus muscle of 6 month old male *Ldb3*^{Ala165Val/+} mice (blue) versus *Ldb3*^{+/+} littermates (red) against stimulation frequencies (Hz). Mean ± SEM: 148 ± 5 mN/mm² versus 187 ± 3 mN/mm²; 160 Hz; n = 4 *Ldb3*^{Ala165Val/+} mice and n = 3 *Ldb3*^{+/+} mice. The Bonferroni's multiple comparisons test significant p-values between groups are shown.

2IV. We modified text in Results section on page 6, **lines 119 – 123** as, “We found that the reduced grip strength was associated with decreased specific isometric force of the extensor digitorum longus muscle in 6 month old *Ldb3*^{Ala165Val/+} mice compared with that in *Ldb3*^{+/+} littermates (Fig. 1e). A two way ANOVA with post hoc Bonferroni correction yielded a significant effect of genotype F(1, 5) = 17.93, p = 0.008.”

3. *A serious issue that first appears in figure 2 but is present for all of the pathology is the complete absence of control images for immunolabelling, histology, and electron microscopy. Control images, of WT littermates must be included for every image shown in figure 2 and subsequent figures.”*

Response 3:

We added *Ldb3*^{+/+} littermate control images in new figures 2, 3, and 4 as shown below.

3I. New Figure 2. Panels a, c, and d show images from *Ldb3*^{+/+} littermates.

Corresponding changes in Results lines 130 –131 “Muscle histology in mutant mice at 4 months of age was similar to that in wildtype mice (Fig. 2a).”; lines 161 – 162, panel c, “Such protein aggregates were not observed in *Ldb3*^{+/+} mice.”; and lines 164 – 165 “...normal Z-disc and sarcomere architecture in their *Ldb3*^{+/+} littermates (Fig. 2d-f).”

Legend lines 994, panel a, “...and 8 month old *Ldb3*^{+/+} littermates (n = 9).”; lines 996 – 997, panel b, “Muscle staining is normal in *Ldb3*^{+/+} mice...”; and lines 1001 – 1004, panel d, “...but not in *Ldb3*^{+/+} mice. d-f, Electron microscopy of the soleus muscle longitudinal section of 6 month old *Ldb3*^{+/+} mice (n =3) shows normal Z-disc (black arrows) and the sarcomere architecture (d),...”

Images of figures 3 and 4 are on next pages.

3II. New Figure 3. Panels a and b show images from *Ldb3*^{+/+} littermates.

Corresponding changes in Results **lines 191 – 192**, panel a: “Such protein aggregates were not seen in wildtype littermates.”; **lines 200 – 201**, panel b: “... but not in their *Ldb3*^{+/+} littermates”

Legend **line 1023**, panel a, “... *Ldb3*^{+/+} littermates (n = 5)...”; **line 1028**, panel b, ... “8 month old *Ldb3*^{+/+} mice (b; n = 6)...”, **lines 1032 – 1033**, panels a-b, “Such protein aggregates are not seen in the sections obtained from *Ldb3*^{+/+} mice.”

3III. New Figure 4. Panels a and f show images from *Ldb3*^{+/+} littermates. (image on next page)

Corresponding changes in

Results **lines 203 – 205**: Serial longitudinal tibialis anterior muscle sections of 4- and 8- month old *Ldb3*^{+/+} mice showed normal Z-disc distribution of LDB3, filamin C, myotilin, BAG3, and HSPA8 proteins (Fig. 4a and f).

Legends **lines 1039 – 1041**, “... and their *Ldb3*^{+/+} littermates. **a**, Muscle sections of 4 month old *Ldb3*^{+/+} mice (n = 4) show normal Z-disc staining for LDB3, filamin C, myotilin, BAG3, and HSPA8 proteins.”; and **lines 1047 – 1049**, “**f**, Muscle sections of 8 month old *Ldb3*^{+/+} mice (n = 5) show normal Z-disc staining for LDB3, filamin C, myotilin, BAG3, and HSPA8 proteins.”

4. *In Figure 2g the statistical comparison is showing the effect of colchicine treatment and should be comparing flux between the WT and mutant mice.”*

Response 4:

We added a dot plot showing statistical comparison of colchicine effect on flux between groups in Fig. 2i as shown below.

Corresponding changes in Results lines 165 – 168, “An *in vivo* “autophagy flux” assay²¹ detected an increase in autophagy markers LC3-II and sequestosome 1 (p62 / SQSTM1) following 3 days of treatment with colchicine in skeletal muscle of 4 month old *Ldb3*^{Ala165Val/+} mice compared to *Ldb3*^{+/+} littermates (Fig. 2g-i).”

Legends lines 1011 – 1015, “i, Dot plot of LC3A/B-II and sequestosome-1 flux [ΔCOL – PBS]. Mean ± SD flux: LC3A/B-II: 8.4 ± 0.8 versus 3.3 ± 0.4 and sequestosome-1: 6.6 ± 0.5 versus 2.3 ± 0.6 in *Ldb3*^{Ala165Val/+} mice and *Ldb3*^{+/+} littermates, respectively... the two-tailed unpaired *t*-test comparison (i) significant p-values are shown.

5. *Figure 2g – in addition to the graph not showing the correct comparison the interpretation does not appear to be correct. The authors conclude the autophagy is dysfunctional, but flux is higher in the mutant mice, instead suggesting that autophagy is increased in the mutant animals. The interpretation of the data needs to be re-examined, and the conclusions altered if necessary.”*

Response 5: We modified the text in Results section line 171, “...increased lysosomal autophagy...”. We discuss interpretation of increased autophagy markers in mice in lines 354 – 358 as, “The aggregation of damaged filamin C aggravates the autophagy pathway in muscle fibers of *Ldb3*^{Ala165Val/+} mice. This is evidenced by marked accumulations of the CASA chaperones in the muscle fibers containing protein aggregates and increased levels of the autophagy markers after colchicine blockage in skeletal muscle of 4 month old mice.”

6. *Line153 ‘Data not shown’. The data should be provided to support all statements in the manuscript. This includes line 168 where it is reported that soleus and vastus were examined, but only the vastus is shown.”*

Response 6:

- 6l. We added the representative heart histology images of wildtype and mutant mice in new Supplementary Fig. 4 as below.

Corresponding changes in Results, **lines 173 – 176**, “Cardiac ventricular muscle fibers of *Ldb3^{Ala165Val/+}* mice displayed normal histology at 9 months of age (Supplementary Fig. 4a-b), which correlates with clinical observations of cardiomyopathy not being a regular feature in patients known to have the same mutation²².”

The legend is in **Supplemental Information** file as, “**Supplementary Fig. 4: Heart histology of *Ldb3^{Ala165Val/+}* mice.** Representative hematoxylin and eosin stained histological sections of heart showing normal morphology in 9 month old *Ldb3^{Ala165Val/+}* mice (n = 3) and *Ldb3^{+/+}* littermates (n = 3). **a**, Low magnification (scale bar = 1mm) of digitally stitched images, showing the interventricular septum (IVS) and ventricular walls (RV, right ventricle; LV, left ventricle). **b**, Transverse section (TS) and longitudinal section (LS) of the left ventricular myocardium at high magnification (scale bar = 50 μm).”

- 6II. We added muscle immunofluorescence images of LDB3 and filamin C stained soleus muscles obtained from 6-month old WT and mutant mice as new Fig. 3a as below.

Corresponding changes in Results, **lines 187 – 189**, “We found filamin C aggregates in muscle fibers that had normal LDB3 immunostaining in transverse vastus and soleus muscle sections of 6 month old *Ldb3^{Ala165Val/+}* mice (Fig. 3a).”; and **lines 191 – 192**, “Such protein aggregates were not seen in wildtype littermates.”

Legend **lines 1021 – 1026**, “**a**, Representative immunofluorescence on ... soleus muscle transverse section of 6 month old *Ldb3^{Ala165Val/+}* mice (n = 9 and 5, respectively) and *Ldb3^{+/+}* littermates (n = 5) ... stained with filamin C and LDB3 antibodies. White arrows indicate muscle fibers with sarcoplasmic filamin C aggregates that have normal sarcoplasmic LDB3 distribution in *Ldb3^{Ala165Val/+}* mice...”

- 6III. We removed “unpublished data” for, “only negligible expression of exon 6 containing LDB3 isoforms is found in mouse heart and nervous system⁹” in Discussion section **line 393 – 395** as the cited reference already provides the data supporting this statement.

7. *Line 184 refers to disintegration occurring in mice at 6 months and older but no disintegration is shown and this, and subsequent statements of ‘disintegration’ need to be rephrased. The timepoint in the text (6 months) does not match the figure (4 and 8 months). The correct age of the mice needs to be checked and the text and figures should match.*”

Response 7: We rephrased “disintegration” throughout the manuscript. We modified/added the age in the text to match that in the corresponding images and also throughout the manuscript.

8. *Line 187. It is reported that CASA component aggregates were not observed in adjoining fibres but the adjoining fibres are not shown and should be included in the images.”*

Response 8: We modified the Figs. 3 and 4 to include adjoining fibers as feasible in the images as shown on page 4 and 5 of this document. And modified text in Result section lines **210 – 213**, “Prominent accumulations of the CASA chaperones were found in muscle fibers harboring filamin C and other myofibrillar protein aggregates but not in adjoining fibers without protein aggregation (Figs. 3c-d and 4d-e, i-j).”

Fig. 3c-d

Fig. 4 d-e and i-j

9. *Line 239 it is reported that CASA functions are impaired, but the function has not been examined and therefore analysis of the turnover of CASA targets needs to be included or the statement rephrased or removed.”* And

10. *“10) Similarly line 240 refers to muscle fibre degeneration but this is not shown in any of the data provided.”*

Response 9 and 10: We rephrased the text in Results section lines **263 – 265** as, “It should be noted that filamin C aggregates were observed in skeletal muscle fibers of 4 month old mutant mice (Fig. 4b,d), whereas Z-disc myofibrillar disruption is evident after 6 months of age (Fig. 2e; Fig. 4g-i)”.

11. *Line 257 The authors report a change in protein levels for PKCalpha and TSC2. However, they conclude that this is responsible for the MFM phenotype. There is no evidence provided to support this conclusion.”*

Response 11: We revised the text in Results lines **280 - 283** as following, “Downregulation of PKC α and TSC2 indicates a likely mechanism for the observed aggregation of damaged filamin C in muscle fibers of *Ldb3*^{A165V/+} mice as PKC α stabilizes filamin C in muscle Z-disc through phosphorylation³⁴, and TSC2 initiates CASA-mediated degradation of damaged filamin through local mTORC1 inhibition³⁵.”

12. *Line 274 it is also concluded that disintegration occurs in fibers with aggregates, but disintegration is not shown. From the data provided no disintegration is apparent and the manuscript needs to be rephrased accordingly.”*

Response 12: We rephrased the statement in Discussion lines **298 – 300** as, “Further, pathology studies show that muscle fibers with protein aggregates develop the Z-disc myofibrillar disruption in skeletal muscle of *Ldb3*^{A165Val/+} mice.”

13. Line 321. It is again concluded that PKC α exquisitely controls Z-disc protein homeostasis. No examination of Z-disc protein turnover or levels is shown and the only examination of PKC α is the level of expression, with no functional analyses. Therefore, this conclusion is also not supported by evidence in the manuscript.”

Response 13: We removed this sentence from Discussion line 346.

“Minor issues”:

1) Line 77 – ‘murine models of MFM have not yet been reported’ – LDB3 needs to be added to this statement as other murine models of MFM have been reported.”

Response minor issue 1: We revised the statement in Introduction line 80 – 81, as, “...the murine models of LDB3-MFM have not yet been reported.”

2) Figure 1 – it would be useful if the legend for the colours was included in the image. The brackets should align with the centre of the groups being compared.”

Response minor issue 2: We revised Fig. 1 by adding the color legend and aligning the brackets as below.

3) The authors should check the approved nomenclature eg HSPB8 not Hsp22 and use throughout.”

Response minor issue 3: We replaced Hsp22 with HSPB8, Hsc70 with HSPA8, and p62 with sequestosome 1 throughout in text and figures.

Reviewer #2 (Remarks to the Author):

1. “On page 6, line 110, it is stated that *Ldb3*^{A165Val/+} mice showed a progressive decline in the grip strength test. However, this is only at 6 months vs. 3 months, while the difference between WT and mutant mice is less at 9 months. It would therefore be preferable to avoid the word “progressive”. Since the mice are compared at different time points, the statistical test should be done with ANOVA with appropriate post hoc analysis and not the unpaired t-test. Also in Figure 1e, ANOVA would be a more appropriate statistical test.”

Response 1:

11. We removed “progressive” from the text in Results line 115, as suggested and rephrased the text as, “Grip strength tests showed a significant decline in the *Ldb3*^{A165Val/+} mice compared with *Ldb3*^{+/+} littermates at 3, 6, and 9 months of age (Fig. 1d).”

1II. We performed two way ANOVA statistical tests with post hoc Bonferroni correction for the data in Figs. 1d and 1e (see tables below).

Fig. 1d

Bonferroni's multiple comparisons test	Mean diff.	95.00% CI of diff.	Significant?	Summary	Adjusted P Value
WT - Mutant					
3 months	1.539	0.4156 to 2.662	Yes	**	0.0040
6 months	3.501	2.391 to 4.610	Yes	****	<0.0001
9 months	1.411	0.1823 to 2.640	Yes	*	0.0191

Fig. 1e

Bonferroni's multiple comparisons test	Mean Diff.	95.00% CI of diff.	Significant?	Summary	Adjusted P Value
WT - Mutant					
20 Hz	12.97	-37.62 to 63.55	No	ns	>0.9999
40 Hz	17.91	-24.86 to 60.68	No	ns	0.6281
60 Hz	28.95	-43.35 to 101.2	No	ns	0.8008
80 Hz	34.67	-12.46 to 81.80	No	ns	0.1619
100 Hz	37.67	-3.161 to 78.49	No	ns	0.0696
120 Hz	39.27	3.851 to 74.70	Yes	*	0.0322
140 Hz	40.07	7.239 to 72.91	Yes	*	0.0214
160 Hz	39.60	9.840 to 69.36	Yes	*	0.0152
180 Hz	36.89	8.226 to 65.56	Yes	*	0.0191

1III. We modified Fig. 1d and e panels and manuscript text as below.

Corresponding changes in Results, **lines 115 – 123**, “Grip strength tests showed a significant decline in the *Ldb3*^{Ala165Val/+} mice compared with *Ldb3*^{+/+} littermates at 3, 6, and 9 months of age (Fig. 1d). A two-way analysis of variance (ANOVA) with post hoc Bonferroni correction yielded significant effects of genotype, $F(1, 56) = 63.3$, $p < 0.0001$ and age, $F(2, 56) = 32.9$, $p < 0.0001$, as well as the age and genotype interaction, $F(2, 56) = 6.5$, $p = 0.003$. We found that the reduced grip strength was associated with decreased specific isometric force of the extensor digitorum longus muscle in 6 month old *Ldb3*^{Ala165Val/+} mice compared with that in *Ldb3*^{+/+} littermates (Fig. 1e). A two way ANOVA with post hoc Bonferroni correction yielded a significant effect of genotype $F(1, 5) = 17.93$, $p = 0.008$.

digitorum longus muscle in 6 month old *Ldb3*^{Ala165Val/+} mice compared with that in *Ldb3*^{+/+} littermates (Fig. 1e). A two way ANOVA with post hoc Bonferroni correction yielded a significant effect of genotype $F(1, 5) = 17.93$, $p = 0.008$.

Legend **lines 971 – 983, d**, Dot plot comparing maximum all paws grip force of five pulls per mouse normalized to body weight (g/g) between $Ldb3^{Ala165Val/+}$ mice (blue) and $Ldb3^{+/+}$ littermates (red) at 3, 6, and 9 months of age. Mean \pm SEM: 9.2 ± 0.3 g/g versus 10.7 ± 0.3 g/g at 3 months, 6.9 ± 0.3 g/g versus 10.4 ± 0.3 g/g at 6 months, and 6.5 ± 0.2 g/g versus 7.9 ± 0.3 g/g at 9 months; $n = 10 - 13$ $Ldb3^{Ala165Val/+}$ mice and $n = 8 - 10$ $Ldb3^{+/+}$ mice per age group. The two-way ANOVA Bonferroni's multiple comparisons test significant p-values between groups are shown. **e**, Dot plot shows maximal isometric force (mN/mm²) generated by the extensor digitorum longus muscle of 6 month old male $Ldb3^{Ala165Val/+}$ mice (blue) versus $Ldb3^{+/+}$ littermates (red) against stimulation frequencies (Hz). Mean \pm SEM: 148 ± 5 mN/mm² versus 187 ± 3 mN/mm²; 160 Hz; $n = 4$ $Ldb3^{Ala165Val/+}$ mice and $n = 3$ $Ldb3^{+/+}$ mice. The two-way ANOVA Bonferroni's multiple comparisons test significant p-values between groups are shown.

2. "For the histological analyses in Figure 2, please state how many mice were analyzed per time point? Also, maybe this information would be more appropriate in the figure legend than in the text (page 6, line 126-127). In Figure 2, pictures of the histological, immunostaining, and ultrastructural analyses are shown only for the mutant mice. Please show corresponding images from WT mice for comparison. The same applies for Figure 3 and 4."

Response 2:

- 2I. We modified the legends by adding the number of mice analyzed for each image throughout the manuscript and removed the number of mice from the Result, as suggested. We also added corresponding data images of WT littermates in Figs. 2, 3 and 4 as below.
- 2II. New Figure 2. Panels a, c, and d show images from $Ldb3^{+/+}$ littermates.

Corresponding changes in Results **lines 130 –131** "Muscle histology in mutant mice at 4

months of age was similar to that in wildtype mice (Fig. 2a)."; **lines 161 – 162**, panel c, "Such protein aggregates were not observed in $Ldb3^{+/+}$ mice."; and **lines 164 – 165** "...normal Z-disc and sarcomere architecture in their $Ldb3^{+/+}$ littermates (Fig. 2d-f)."

Legend **lines 994**, panel a, "...and 8 month old $Ldb3^{+/+}$ littermates ($n = 9$)."; **lines 996 – 997**, panel b, "Muscle staining is normal in $Ldb3^{+/+}$ mice..."; and **lines 1001 – 1004**, panel d, "...but not in $Ldb3^{+/+}$ mice. d-f, Electron microscopy of the soleus muscle longitudinal section of 6 month old $Ldb3^{+/+}$ mice ($n = 3$) shows normal Z-disc (black arrows) and the sarcomere architecture (d),..."

2III. New Figure 3. Panels a and b show images from *Ldb3*^{+/+} littermates.

Corresponding changes in Results **Lines 191 – 192**, panel a: “Such protein aggregates were not seen in wildtype littermates.”; **lines 200 – 201**, panel b: “... but not in their *Ldb3*^{+/+} littermates”

Legend **line 1023**, panel a, “... *Ldb3*^{+/+} littermates (n = 5)...”; **line 1028**, panel b, ... “8 month old *Ldb3*^{+/+} mice (b; n = 6)...”, **lines 1032 – 1033**, panels a-b, “Such protein aggregates are not seen in the sections obtained from *Ldb3*^{+/+} mice.”

2IV. New Figure 4. Panels a and f show images from *Ldb3*^{+/+} littermates. (image on next page)

Corresponding changes in

Results **lines 203 – 205**: Serial longitudinal tibialis anterior muscle sections of 4- and 8- month old *Ldb3*^{+/+} mice showed normal Z-disc distribution of LDB3, filamin C, myotilin, BAG3, and HSPA8 proteins (Fig. 4a and f).

Legend **lines 1039 – 1041**, “... and their *Ldb3*^{+/+} littermates. **a**, Muscle sections of 4 month old *Ldb3*^{+/+} mice (n = 4) show normal Z-disc staining for LDB3, filamin C, myotilin, BAG3, and HSPA8 proteins.”; and **lines 1047 – 1049**, “**f**, Muscle sections of 8 month old *Ldb3*^{+/+} mice (n = 5) show normal Z-disc staining for LDB3, filamin C, myotilin, BAG3, and HSPA8 proteins.”

3. “In the immunofluorescence analyses in Figure 2c-d, would it be possible to quantify the percentage of cells showing sarcoplasmic accumulation of Z-disc proteins and ubiquitin in mutant mice compared to WT?”

Response 3: We added quantitation of percentage muscle fibers containing filamin C and ubiquitin in the soleus muscle sections of 8 month old mutant mice in Supplementary Fig. 3a shown below.

Corresponding changes in Result lines 156 – 162, “Counts of a mean of 498 fibers in transverse soleus muscle section of five mutant mice each showed a mean of 64% and 44% muscle fibers contained filamin C and ubiquitin accumulations, respectively (Supplementary Fig. 3a). These protein accumulations occurred in multiple or diffuse form in muscle sarcoplasm. The abnormal fibers were distributed focally surrounded by muscle fibers without protein aggregation. Such protein aggregates were not observed in *Ldb3*^{+/+} mice.”

The legend is in **Supplemental Information** file as, “**Supplementary Fig. 3: Quantitation of percentage skeletal muscle fibers with sarcoplasmic filamin C and ubiquitin accumulations in *Ldb3*^{Ala165Val/+} mice.** a, Representative immunofluorescence images of the soleus muscle transverse section obtained from 8 month old *Ldb3*^{Ala165Val/+} mice and *Ldb3*^{+/+} mice (n = 5 each) stained with filamin C and ubiquitin antibodies, which were used for quantitation of percentage fibers with protein aggregates. Arrows show examples of counted abnormal fibers containing protein accumulations in mutant mice. Note focal cluster of abnormal fibers is surrounded by fibers without any aggregates. Bar-dot plots show quantitation of percentage muscle fibers containing filamin C and ubiquitin accumulations in the soleus muscle of mutant mice.”

4. “On page 7, line 138, it is stated that “Non-specific myopathic changes such as variability in fiber size, increases in internal nuclei, rounding of fibers, and increased in interstitial space were also observed”. However, no data are shown. The variability in fiber size could be due to altered fiber type distribution and/or fiber type specific fiber size. Therefore please provide quantification of fiber size and fiber type distribution, including fiber type-specific fiber size. It would also be useful to show images of the other mentioned myopathic changes, when possible with quantification, at least in the supplemental file.”

Response 4:

We added images Fig. 2b and Supplementary Fig. 2.

Fig. 2b. Corresponding changes in Result lines 136 – 139, ““In addition, non-specific myopathic changes such as increases in internal nuclei, hypertrophic fibers, and muscle fibers with rounded contour were observed in the vastus and tibialis anterior muscles of mutant mice (Fig. 2b).”

Legend lines 994 – 999, b, Representative images are GT – stained transverse frozen vastus and tibialis anterior muscle sections of 8 month old *Ldb3*^{Ala165Val/+} mice (n = 16). Muscle staining is normal in *Ldb3*^{+/+} mice and 4 month-old *Ldb3*^{Ala165Val/+} mice but shows ... muscle fiber with rounded contour, hypertrophied fiber, and internal nuclei (b) in 8 month old *Ldb3*^{Ala165Val/+} mice.”

Supplementary Fig. 2. Corresponding changes in Result lines 139 – 152, “To quantitate the changes in myonuclear location, we performed morphometry with wheat germ agglutinin (WGA) to outline muscle membrane and distinguish muscle nuclei. Relative to wildtype littermates, *Ldb3*^{Ala165Val/+} mice had a significantly higher percentage of fibers with internal nuclei in the tibialis anterior muscle (p < 0.01; Supplementary Fig. 2a). Muscle fiber type distribution was unchanged in the soleus and vastus muscles of *Ldb3*^{Ala165Val/+} mice (Supplementary Fig. 2b-d). Muscle fiber size distribution for types I and IIA in *Ldb3*^{Ala165Val/+} mice was shifted towards right compared to those from *Ldb3*^{+/+} mice, whereas distribution of type IIB fiber size was unchanged (Supplementary Fig. 2e). Mean minimal Feret’s diameter for fiber types I and IIA was higher in *Ldb3*^{Ala165Val/+} mice relative to wildtype mice, but the change was not statistically significant in the Bonferroni’s multiple comparison test (Supplementary Fig. 2f). Accordingly, variance coefficient of the minimal muscle fiber diameter was unchanged for all fiber types in *Ldb3*^{Ala165Val/+}

mice (Supplementary Fig. 2g).”

The legend is in **Supplemental Information** file as, “Supplementary Fig. 2: Quantitation of percentage internal nuclei, fiber type, and fiber-type specific size in skeletal muscle of *Ldb3*^{Ala165Val/+} mice. a, Immunofluorescence of the tibialis anterior muscle transverse section of 8 month old *Ldb3*^{Ala165Val/+} mice (n = 4) and *Ldb3*^{+/+} mice (n = 3) labeled with wheat germ agglutinin (WGA; muscle membrane; green) and DAPI (nuclei; magenta). The rectangles in insets define the boundary of main images. Percent fibers with internal nuclei in *Ldb3*^{Ala165Val/+} mice (blue; Mean ± SEM: 6.6% ± 0.7%) and *Ldb3*^{+/+} mice (red; Mean ± SEM: 1.1% ± 0.4%) are shown in bar-dot plot representing a mean of 764 fibers per muscle (Unpaired two-tailed *t*-test p < 0.01). b-c, Immunofluorescence of the soleus (b) and vastus (c) muscle transverse section of 8 month old *Ldb3*^{Ala165Val/+} mice (n = 4 and 3, respectively) and *Ldb3*^{+/+} mice (n = 3 each) labeled with myosin heavy chain types I, IIA, IIB, and IIX antibodies for fiber typing and WGA for muscle fiber size (minimal Feret’s diameter) measurement. d, Fiber type percentages are shown for *Ldb3*^{Ala165Val/+} mice (blue) and *Ldb3*^{+/+} mice (red) in bar-dot plots. The data represent a mean of 615 fibers and 436 fibers per muscle, in the soleus and vastus muscles, respectively. e, Histogram of the fiber-type specific size in *Ldb3*^{Ala165Val/+} mice (blue) and *Ldb3*^{+/+} mice (red) are

shown. The data represent a mean of 100 type I fibers, 141 type IIA fibers, and 288 type IIB fibers per muscle, and n = 4 mice for types I and IIA fibers and n = 3 mice for type IIB fibers. f, The mean minimal Feret's diameter of fiber types in *Ldb3^{Ala165Val/+}* mice (blue) and *Ldb3^{+/+}* mice (red) are shown in bar-dot plot (Mean ± SEM: type I, 40.4 mm ± 0.4 mm versus 37.2 mm ± 0.4 mm ; type IIA, 39.1 mm ± 0.3 mm versus 34.3 mm ± 0.3 mm; and type IIB, 45.1 mm ± 0.4 mm versus 45 mm ± 0.4 mm). The two way ANOVA Bonferroni's multiple comparison test p > 0.05. g, The variance coefficient (VC) of the muscle fiber diameter, calculated as 1000 x SD / Mean, in both groups are shown in bar-dot plot (Mean ± SEM: type I, 182.7 ± 8.4 vs 204.1 ± 8.5; type IIA, 189.9 ± 7.9 versus 187.2 ± 12.6; and type IIB 269.6 ± 14.8 versus 264.3 ± 15.6). The bar and error bar in plots represent Mean and SEM, respectively. Scale bars = 100 μm (insets 500 μm).”

5. a. “In figure 2g, please show the individual data points on the graphs with the densitometric analysis.

Response 5a: The individual data points representing Mean values of triplicate assays ran on muscle lysates obtained from mice (n = 3 per treatment group) are shown in new Fig. 2h as below.

Fig. 2h. Legend lines 1007 – 1011, g-h, Immunoblotting analysis (blot and dot plot) of LC3A/B-II and sequestosome-1 (p62) protein levels, relative to GAPDH, in the vastus muscle of 4 month old *Ldb3^{Ala165Val/+}* and *Ldb3^{+/+}* littermates after three days of colchicine (COL) or PBS treatment. n = 3 mice per

treatment group and triplicate assays.

5b. From the figure it appears as if the samples from WT and mutant mice were run on separate gels, although the main point of the analysis should be to compare protein levels in WT and mutant mice.

Response 5b:

We added the following text to Methods lines 705 – 710, “The samples were run on two gels (one mini gel could not accommodate all sample loadings) in the same electrophoresis tank. The protein transfer from both gels was done simultaneously on a single membrane which was then processed for immunoblotting. For densitometric analysis, all the band intensities were normalized to that of one PBS-treated *Ldb3^{+/+}* sample in each membrane. See Supplementary Fig. 6b for full-length blot images.”

5c. Direct comparison of protein levels in WT vs. mutant mice treated or not with colchicine should be performed by ANOVA by running the samples from WT and mutant mice on the same gel.

Response 5c: We used a two-way ANOVA multiple comparison test with post hoc Bonferroni correction (Test tables below), which is represented in Fig. 2h.

LC3-II:

Bonferroni's multiple comparisons test	Mean Diff.	95.00% CI of diff.	Significant?	Summary	Adjusted P Value
Colchicine - PBS					
WT	3.340	5.128 to 1.552	Yes	**	0.0018
A165V/+	8.440	10.23 to 6.652	Yes	****	<0.0001

Sequestosome-1 (p62):

Bonferroni's multiple comparisons test	Mean Diff.	95.00% CI of diff.	Significant?	Summary	Adjusted P Value
Colchicine - PBS					
WT	2.326	3.815 to 0.8368	Yes	**	0.0052
A165V/+	6.590	8.079 to 5.101	Yes	****	<0.0001

5d. For LC3, please quantify the LCII/LC3I ratio, which is a more appropriate measure for the autophagic state.”

Response 5d: We corrected the antibody name in Supplementary Table 4 from LC3 I/II to LC3 A/B antibody. The LC3A/B antibody (#12741; Cell Signaling Technology) consistently showed stronger signal for LC3-II compared to LC3-I. Direct comparison of relative affinity between LC3-II and LC3-I for this and other LC3 antibodies has not been done as per the manufacturer. Evaluation of LC3-II levels using a loading control in the presence and absence of lysosomal autophagy inhibitors (colchicine) as a measure of autophagy flux in mouse skeletal muscle has been published previously (cited as reference 21 in manuscript). Cellular LC3-II levels using a loading control (in preference to LC3-I) as autophagosome status marker has been discussed in review papers (Rubinsztein D.C., et al Autophagy 2009; 5:585-589, <https://doi.org/10.4161/auto.5.5.8823>; Mizushima N and Yoshimori T, Autophagy 2007; 6: 542-545, <https://doi.org/10.4161/auto.4600>). For these reasons, the levels of LC3-II relative to GAPDH loading control were compared among samples.

6. “In Figure 3a, please show the WT control and quantify the amount of fibers with filamin C accumulation in mutant mice compared to WT. Please provide the n for both mutant and WT mice analyzed.”

Response 6:

6l. We added images of WT littermates in Fig. 3a and modified the legend by adding the number of mice analyzed.

Corresponding change in Result lines 191 – 192, “Such protein aggregates were not seen in wildtype littermates.”

Legend Fig. 3a lines 1021 – 1023, “a, Representative immunofluorescence on frozen vastus lateralis and soleus muscle transverse section of 6 month old *Ldb3^{Ala165Val/+}* mice (n = 9 and 5, respectively) and *Ldb3^{+/+}* littermates (n = 5).”

6II. We performed quantitation of percentage muscle fibers containing filamin C accumulations in the vastus muscle of mutant mice.

The following text is added in Results lines 189 – 192, “Counts of a mean of 623 fibers from vastus muscle transverse sections of five mutant mice each showed a mean of 15% fibers contained filamin C accumulations (Supplementary Fig. 3b). Such protein aggregates were not seen in wildtype littermates.”

The legend is in **Supplemental Information** file as, “**Supplementary Fig. 3: Quantitation of percentage skeletal muscle fibers with**

sarcoplasmic filamin C and ubiquitin accumulations in *Ldb3^{Ala165Val/+}* mice. b, Representative immunofluorescence images of the vastus muscle transverse section obtained from 6 month old *Ldb3^{Ala165Val/+}* mice and *Ldb3^{+/+}* mice (n = 5 each) stained with filamin C antibody, which were used for quantitation of percent fibers with protein aggregates. Bar-dot plot shows quantitation of percentage muscle fibers containing filamin C accumulations in the vastus muscle of mutant mice. Muscle fibers with protein aggregates are not seen in the soleus and vastus muscles of *Ldb3^{+/+}* mice. The data represent a mean of 498 fibers and 623 fibers per muscle, for the soleus and vastus muscles, respectively. The bar and error bar in plots represent Mean and SEM, respectively. Scale bars = 100 μ m.”

7. “In Figure 6, based on the trend towards reduced TSC2 levels at 8 months of age in the RPPA analysis, it would be preferable to show Western blot analysis for TSC2 also at 8 months of age.”

Response 7: We added Western blot analysis for TSC2 at 8 months of age in Fig. 6e-f.

Corresponding change in Results lines 277 – 279, “Whole muscle levels of PKC α in 4 and 8 month old *Ldb3^{Ala165Val/+}* mice were decreased to about 50% of those in *Ldb3^{+/+}* littermates, and levels of TSC2 in the mutant mice were reduced to almost two-thirds of those in *Ldb3^{+/+}* mice (Fig. 6c-f).”

Legend lines 1097 – 1099, e-f, Immunoblotting analysis (blot and dot plot) for protein levels of PKC α and TSC2 relative to vinculin in the vastus

muscle of 8 month old *Ldb3*^{Ala165Val/+} mice (n = 4 and 5, respectively) and *Ldb3*^{+/+} littermates (n = 3 and 4, respectively).

8. *“It is stated that gender-matched littermates were used for the experiments but not whether preferably males or females were used for the analyses. Were the analyses done on mixed females and males? If so, were there any differences in for example grip strength and isometric force between genders and were the differences between WT and mutant mice similar in males and females?”*

Response 8: Males were used for muscle strength analysis. We observed intra- and inter-group variability in strength measurements of female mice and therefore did not include female mice in the analysis. We have added the following text in Fig. 1 legend **lines 986 – 987**, “Male mice were used in these tests as female mice showed intra- and inter-group variability.” For all other data we found no difference between sexes, therefore gender-matched mixed females and males were included.

“Minor issue”:

9. *“Page 8, line 163: “Patients usually present in advanced stages of myopathic disease” should be corrected to “Patients usually present with...”*

Response 9: We have rephrased the sentence in Results **lines 184 – 185**, “Patients usually present with advanced stages of myopathic disease at the time of biopsy, thereby limiting studies of early pathological features in the MFM.”

10. *“Page 8, line 168: It is unclear what is meant with the statement: “The filamin C antibody staining was more strongly positive compared with LDB3 antibody in the vastus lateralis muscle fibers of patients sharing the same mutation”. The intensity of the two different antibodies cannot be directly compared. Please modify.”*

Response 10: We have revised the sentence in Results **lines 192 – 194**, “The filamin C accumulations were more prominent compared to LDB3 in the vastus lateralis muscle fibers of patients sharing the same mutation (Fig. 3a).”

11. *“In Figure 5a, please write the number of replicates and how many times the assay was repeated.”*

Response 11: We added the text, “Data represent n = 5 mice per group and triplicate assays.” in the legend for Fig. 5a in **line 1063**.

Reviewer #3 (Remarks to the Author):

1. *“The variant nomenclature should follow HGVS recommendations (<https://varnomen.hgvs.org/recommendations/>), especially:*

A: *“- Protein descriptions should consistently use three-letter amino acid code.”*

Response 1A: We have replaced A165V with Ala165Val throughout the text.

B: *“- Predicted consequences, when only DNA sequence (not RNA or protein sequence) was analysed should be given in parentheses, e.g. NP_001073583.1:(p.Ala165Val).”*

Response 1B: We revised the Results **lines 101 – 103**, to the following text: “The presence of the NP_001034164.1:(p.Ala165Val) mutation was further validated by Sanger sequencing (Fig. 1b).”

C “- Genetic variants should be at least once described in relation to an accepted reference sequence, e.g. NM_001080114.2:c.494C>T, NP_001073583.1:(p.Ala165Val). Please provide accession numbers of the transcripts and protein sequences. The HGVS recommendations apply also to mouse variant.”

Response 1C: We revised the text in Introduction **lines 50 – 53** to the following text: “The dominant p.Ala165Val mutation in exon 6 of *LIM domain-binding protein 3* gene (*LDB3*; HGNC 15710; rs121908334; NM_001080114.2:c.494C>T, NP_001073583.1:(p.Ala165Val)) has been reported in several unrelated families of European ancestry^{6,7}.”

2. “It is worth mentioning that LDB3 p.Ala165Val mutation is likely benign according to ACMG criteria. <https://varsome.com/variant/hg38/rs121908334> Obviously phenotype and epidemiological data clearly indicates that the variant is in fact disease-causing.”

Response 2: We have added the following text in Introduction **lines 53 – 57**: “Whereas this mutation is interpreted as likely benign (<https://varsome.com/variant/hg38/rs121908334>), its penetrance in the long-studied Markesbery-Griggs pedigree is 100% by age 60 years and molecular studies of six unrelated families indicated a founder mutation with a common ancient ancestry⁶.”

3. “Please note that several murine models of MFM were already reported. (Batonnet-Pichon et al. *Myofibrillar Myopathies: New Perspectives from Animal Models to Potential Therapeutic Approaches*. *Journal of neuromuscular diseases*. 2017)”

Response 3: As suggested, we have revised the statement in Introduction **lines 80 – 81** as, “...the murine models of LDB3-MFM have not yet been reported.”

We hope that we have satisfied yours as well as the Reviewer’s concerns and that you would provide us another opportunity to address further concerns.

Thank you for your consideration of our manuscript. We appreciate your time and look forward to your response.

Sincerely,

Ami Mankodi on behalf of all authors

Corresponding author contact information:
35 Convent Drive, Building 35, Room 2A-1002,
Bethesda, MD 20892-3075.
Phone: (301) 827-6690, Fax: (301) 480-3365,
Ami.Mankodi@nih.gov

REVIEWERS' COMMENTS:

Reviewer #1 (Remarks to the Author):

The authors have greatly improved the manuscripts addressing all of the comments appropriately. The data is well presented, and the conclusions are now appropriate for the data. As a result of the changes the manuscript presents a very thorough characterisation of a mouse LDB3 myofibrillar myopathy model and greatly advances the mechanistic understanding of this disease.

In addition to the very high quality improvements to the manuscript I am also grateful to the authors for the excellent response to reviewers. This is the best response I have seen.

The only minor comments I have is that the inclusion of genotypes to figure 4, as in figure 3, would greatly aid the reader, if it was possible. Additionally, the title remains definitive as to the mechanism and, similar the changes made in the discussion, should be modified to reflect the correlative, rather than definitive, evidence.

Reviewer #2 (Remarks to the Author):

The authors have appropriately addressed all my concerns, which has considerably improved the manuscript.

29 January 2021

We thank the reviewers for their critical evaluation of our work. We appreciate their support for the revised version of the manuscript.

Reviewer #1 (Remarks to the Author):

The authors have greatly improved the manuscripts addressing all of the comments appropriately. The data is well presented, and the conclusions are now appropriate for the data. As a result of the changes the manuscript presents a very thorough characterisation of a mouse LDB3 myofibrillar myopathy model and greatly advances the mechanistic understanding of this disease.

In addition to the very high quality improvements to the manuscript I am also grateful to the authors for the excellent response to reviewers. This is the best response I have seen.

The only minor comments I have is that the inclusion of genotypes to figure 4, as in figure 3, would greatly aid the reader, if it was possible. Additionally, the title remains definitive as to the mechanism and, similar the changes made in the discussion, should be modified to reflect the correlative, rather than definitive, evidence.

Response: We thank the reviewer for help in making this work stronger and supporting the manuscript. We have added genotypes in panels of figure 4 and the revised file has been uploaded online. We wish to proceed with the same title for our manuscript.

Reviewer #2 (Remarks to the Author):

The authors have appropriately addressed all my concerns, which has considerably improved the manuscript.

Response: We thank the reviewer for help in making this work stronger and supporting the manuscript.